# Evidence-based modeling of combination control on Kenyan youth HIV/AIDS dynamics

**Marilyn Ronoh**[1]◉*, **Faraimunashe Chirove**[2]◉, **Josephine Wairimu**[1]◉, **Wandera Ogana**[1]◉

**1** School of Mathematics, University of Nairobi, Nairobi, Kenya, **2** Department of Mathematics and Applied Mathematics, University of Johannesburg, Johannesburg, South Africa

◉ These authors contributed equally to this work.
* mcronoh1@gmail.com

**Data Availability Statement:** The data used in this study is owned by the Kenya National Bureau of Statistics and cannot be shared publicly. However, other researchers can obtain access to the data via

## Abstract

We formulate a sex-structured deterministic model to study the effects of varying HIV testing rates, condom use rates and ART adherence rates among Adolescent Girls and Young Women (AGYW) and, Adolescent Boys and Young Men (ABYM) populations in Kenya. Attitudes influencing the Kenyan youth HIV/AIDS control measures both positively and negatively were considered. Using the 2012 Kenya AIDS Indicator Survey (KAIS) microdata we constructed our model, which we fitted to the UNAIDS-Kenya youth prevalence estimates to understand factors influencing Kenyan youth HIV/AIDS prevalence trends. While highly efficacious combination control approach significantly reduces HIV/AIDS prevalence rates among the youth, the disease remains endemic provided infected unaware sexual interactions persist. Disproportional gender-wise attitudes towards HIV/AIDS control measures play a key role in reducing the Kenyan youth HIV/AIDS prevalence trends. The female youth HIV/AIDS prevalence trend seems to be directly linked to increased male infectivity with decreased female infectivity while the male youth prevalence trend seems to be directly associated with increased female infectivity and reduced male infectivity.

## 1 Introduction

Kenya's HIV epidemic ranks fourth worldwide with its general population affected most alongside risk groups such as sex workers, people who inject drugs, men who have sex with men and the youth population [1, 2]. Two decades of successful combination control measures such as HIV testing, public health education campaigns, condom usage, antiretroviral therapy (ART) among others has resulted in the country's significant reduction of the HIV/AIDS prevalence from 10.5% in 1996 to 5.9% in 2015 [3].

Integral to the ongoing fight against HIV/AIDS in Kenya is the component of HIV Counseling and Testing (HCT) with the Government of Kenya and International Development Partners substantially increasing voluntary counseling and testing (VCT) services in the country [4]. Under the Adolescent Reproductive Health Development policy in the 2005-2015 Plan of Action the Government of Kenya sought to establish adolescent friendly voluntary counseling and testing services in a bid to improve and promote accessibility of youth friendly

the National Data Archive (KeNADA) (http://54.213.
151.253/nada/index.php/catalog/94/accesspolicy).

**Funding:** The authors received no specific funding
for this work.

**Competing interests:** The authors have declared
that no competing interests exist.

sexual and reproductive health services [5]. Scale up in innovative approaches to HIV testing in the country include community based HIV testing, door to door testing campaigns and self-testing kits [6, 7]. Despite these great progress in increasing HIV testing centers and new approaches to HIV testing, combined effects of inadequate health services, poverty, sociodemographic characteristics, HIV testing behavior, difficult socio-cultural and psycho-social conditions heavily impact the youth volunteering to HIV testing [8–10]. There is significant gender disparity in factors associated with HIV testing among the youth in Kenya with pregnant female youth required to test for HIV/AIDS due to advanced prevention of mother-to-child transmission(PMTCT) in the country compared to their male counterparts leading to female youth reporting higher HIV testing rates in comparison to male youth of a similar age cohort [2, 3, 11].

The youth in Kenya often engage in unprotected and unplanned sexual intercourse often resulting in sexually transmitted infections, pregnancies and HIV infections [3, 11–13]. While condom use offers dual protection against unplanned pregnancies and protection against HIV/AIDS infection, there is increasing decline in condom use among the youth in Kenya [11, 13]. Some of the factors influencing condom use among the Kenyan youth include perceived individual's risk, peer influence, partner betrayal and socio-cultural factors such as religion, communities, schools and families [3, 12–15]. The youth are easily influenced with their peers negative attitudes to condom use with male peers highly affected compared to female peers [16, 17]. Incorrect use of condoms in these population group places them at a higher risk of HIV/AIDS infection as many of them are experimenting with sex or under the influence of drugs or alcohol [12, 13]. While condom use among the youth remains inconsistent, condom use is generally higher among male youth compared to female youth due to the patriarchal society in Kenya where the male condom is the most preferred method with female youth reporting pressure from male partner not to use condoms [12, 13, 15]. External funding was responsible for most of the free condoms distribution in Kenya and cuts in donor funding has affected majority of the sexually active youth in Kenya who cannot afford to purchase condoms. [18].

Universal Test and Treat strategy by the World Health Organization (WHO) requires that all persons testing positive for HIV/AIDS be initiated on ART immediately irrespective of their CD4+ T cell count so as to achieve 90% diagnosis of all HIV positive persons with 90% of those positively diagnosed initiated on ART so as to achieve 90% viral load suppression [19]. Unfortunately, the adherence rates to ART is proving to be an uphill task among the youth in Kenya [20]. Factors influencing non adherence to ART among the youth in Kenya include stigma associated with disclosure of HIV/AIDS status, lack of adequate support from primary care givers and health workers, treatment fatigue, lack of adequate support structures in schools for youth living with HIV/AIDS, confidentiality breaches by health providers leading to disclosure of patients status to the community, fear of gossip and ridicule, financial constraints leading to failure to honor medical appointments or collect ART drugs and physical and emotional violence meted to orphaned perinatally infected youth by their care givers prompting them to fend for themselves or forcing them to street life [9, 10, 20, 21].

In Kenya, changing key HIV/AIDS control measures among the youth like HIV testing, condom use and ART adherence has faced significant challenges mostly due to societal attitudes towards the uptake of these control measures by the youth [22]. There is significant disparity in societal attitudes by gender towards the youth using some of these HIV/AIDS control measures [22]. On one hand, community norms and structural barriers directly affect condom use among the youth in Kenya with some communities advocating harsh punishment towards the youth using condoms [3, 12–15, 22]. On the other hand, HIV knowledge, HIV-related stigma, income and social support from family and religious affiliations, mental health

(depression, anxiety, stress) and substance use directly affect HIV test-seeking and treatment adherence among the youth [23–25]. These social drivers directly influencing HIV testing, condom use and ART adherence are rarely addressed in mathematical modelling.

Models formulated for HIV/AIDS dynamics have so far informed strategic planning, implementation and evaluation of control programs [26–30]. As of 2000, HIV/AIDS models have coupled interventions such as screening, anti-retroviral therapy (ART) treatment, Prep uptake and condom use [31–37]. Few of these models considered combination control strategies [38]. Real epidemiological data was used in [38–43] to predict HIV/AIDS prevalence subject to the considered control measures.

We seek to show the effects of varying HIV testing rates, condom use rates and antiretroviral adherence rates on the sex-structured AGYW/ABYM disease dynamics in Kenya subject to attitudes influencing disease control such as psycho-social conditions, sociodemographic and socio-cultural characteristics described earlier. In this study, the positive/negative attitudes towards the use of HIV/AIDS control measures are designed to allow HIV testing, condom use and ART adherence to change over time. Using the 2019 UNAIDS-Kenya HIV Surveillance data we fit the AGYW/ABYM model prevalence under the three combination control measures to their respective prevalence data for reliable prevalence predictions and model parameter estimation. HIV/AIDS prevalence among the Adolescent Girls and Young Women (AGYW) population aged 15-24 is high at 5.7% whereas the Adolescent Boys and Young Men (ABYM) population is low at 2.2% [2]. About 73.6% of adolescent girls and young women aged 15-24 tested for HIV/AIDS in 2015 [2]. Similarly, 56% of adolescent boys aged and young men aged 15-24 reported to have tested for HIV/AIDS that year [2]. Approximately 89% of the AGYW reported not using condoms in trusted sexual relations whereas 57.6% of ABYM used condoms at their first sexual encounter [2]. Out of the 268, 586 youth living with HIV/AIDS, 16% are yet to access anti-retroviral therapy (ART) [3]. This model formulation provides a low cost approach to identify key areas for intervention in the real world that could help in reducing new HIV/AIDS infections among the youth in Kenya.

## 2 Methods

### 2.1 Data description

This section details the 2012 Kenya AIDS Indicator Survey description which was used to inform the model formulation described in section 2.2 and the UNAIDS-Kenya National Survey prevalence data description used for the model prevalence fit given in section 3.2.

**2.1.1 Kenya AIDS Indicator Survey (KAIS) data description.** We used the 2012 Kenya AIDS Indicator Survey (KAIS) micro-data obtained from the Kenya National Bureau of Statistics website [44] to construct our model as it included data on HIV testing, sexual behavior and HIV care and treatment of children and adults. Given our interest in HIV testing, sexual behavior and HIV care and treatment of youth, we concentrated only on the all adults and sexual partners data sets. The all adults data set comprised of adolescents and adults aged 15-64 years totaling to 10, 811 with 5,211 males and 5,600 females. The sex partner data set had information regarding sex partner's gender, sexual behavior and HIV/AIDS status. We considered the sex partner data set as we were interested in heterosexual partners. We combined the all adults data set with the sex partners data set and extracted the youth aged 15-24 years. Thus, the combined data set comprised of 3,278 sexually active youth aged 15-24 years with 1,597 ABYM and 1,681 AGYW.

We generated a new variable for HIV/AIDS status knowledge from the combined data set based on HIV testing and it's structure included uninfected unaware, uninfected aware, infected unaware and infected aware. The self-reported status referred to the respondents

self-reported HIV status whereas KAIS confirmed HIV status referred to the respondents HIV status based on laboratory results from the survey [44]. The KAIS confirmed HIV status took into account the viral load testing which we compared to the self-reported status thus adjusting the HIV/AIDS status knowledge of the youth [44]. Uninfected aware population comprised of individuals who reported negative HIV/AIDS status and were KAIS confirmed negative and those who reported negative having tested for HIV/AIDS elsewhere. Uninfected unaware were individuals who reported never tested for HIV/AIDS and were KAIS confirmed negative and those who reported positive HIV/AIDS status and were KAIS confirmed negative. Infected aware included those AGYW / ABYM who reported positive HIV/AIDS status and were KAIS confirmed positive and those who self-reported positive having tested for HIV/AIDS else-where. We classified the infected unaware as those who were HIV infected but reported negative and those who reported never tested for HIV/AIDS. Fig 1(a) and 1(b) gives the data summary for participant gender HIV status knowledge of the youth. HIV/AIDS status knowledge is highest among AGYW at 53.7% and 56.5% among susceptible and infected AGYW in comparison to ABYM. This is consistent with literature findings described in section 1 (see Fig 1(a) and 1(b)). Infected unaware youth are 38.6% more compared to infected aware youth. (see Fig 1(c)).

The question around the use of condom every time with sexual partner was used to determine condom use patterns among the youth and this was tabulated against their HIV status knowledge [44]. Fig 2(a) and 2(b) gives the data summary for participant gender condom use patterns with the youth sexual partners.

Consistent condom use patterns among the uninfected aware ABYM is 18.2% higher in comparison to uninfected aware AGYW (see Fig 2(a)). However, most of the uninfected aware youth fail to use condoms consistently with sexual partners with uninfected aware AGYW ranking highest at 89.6% (see Fig 2(a)). Infected unaware youth inconsistent condom use with sexual partners is 69.2% higher compared to infected aware AGYW/ABYM populations (see Fig 2(c)).

On ART adherence, the questions around currently using ART and daily ART usage were used to determine ART adherence among the infected AGYW/ABYM and this was also tabulated against their HIV status knowledge [44]. Fig 3(a) and 3(b) gives the data summary for participant gender HIV status knowledge and ART usage.

About 38.5% and 30% of infected aware AGYW and ABYM are on ART (see Fig 3(a)). Fig 3(b) shows AGYW/ABYM initiated on ART with daily use, which implies adherence to ART. However, 61.5% and 70% of the infected aware AGYW and ABYM are yet to be initiated on ART (see Fig 3(a)).

**2.1.2 UNAIDS-KENYA HIV surveillance data description.** The National AIDS Control Council in Kenya partners with Avenir Health, UNAIDS, public health professionals, demographers, global epidemiologists and monitoring and evaluation experts to annually provide Kenya's HIV/AIDS estimates [45, 46]. These experts use the Spectrum tools endorsed by UNAIDS to provide these estimates which are based on data from five national surveys (2003 Kenya Demographic and Health Survey, 2007 Kenya AIDS Indicator Survey, 2008/2009 Kenya Demographic and Health Survey, 2012 Kenya AIDS Indicator Survey and 2014 Kenya Demographic and Health Survey) and, data from HIV Sentinel Surveillance among pregnant women, national census and data from various programmes [45]. Hence, Kenya's annual HIV/AIDS prevalence estimates provided by UNAIDS reflect the existing HIV epidemic in the country [45]. For this reason we use the UNAIDS-Kenya HIV Surveillance data on Kenyan youth prevalence to fit the model prevalence for AGYW and ABYM populations. The model fit was also used to estimate the best parameter estimates for some of the model parameters and predict the AGYW and ABYM prevalence for the years 2019—2023. Tables 1–3 give the

(a)

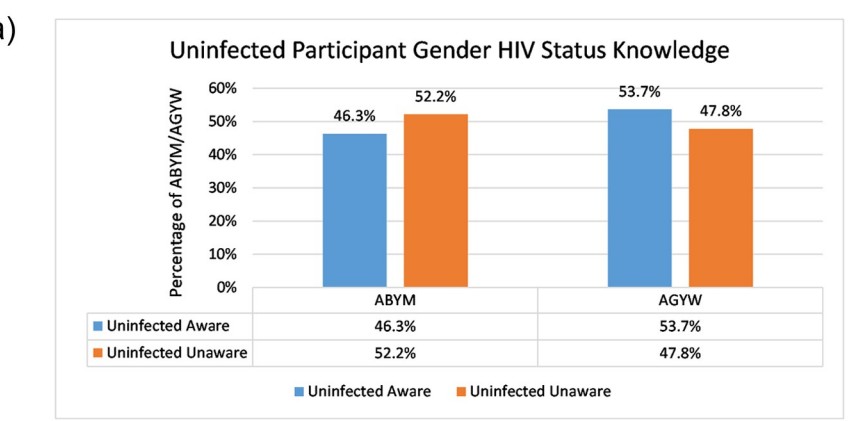

(b)

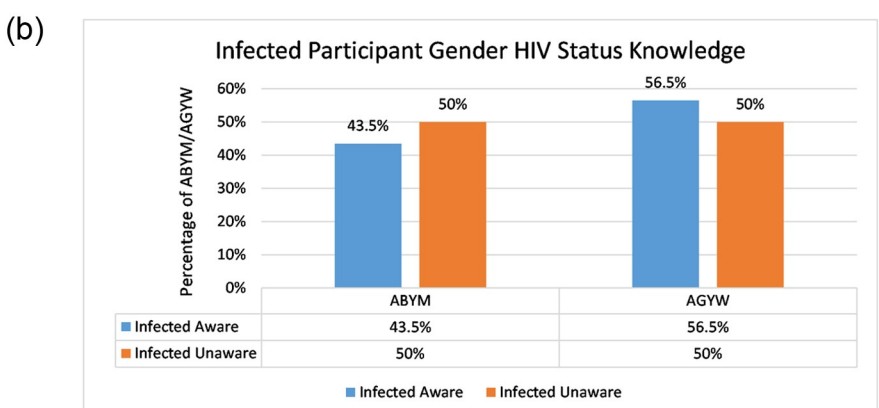

(c)

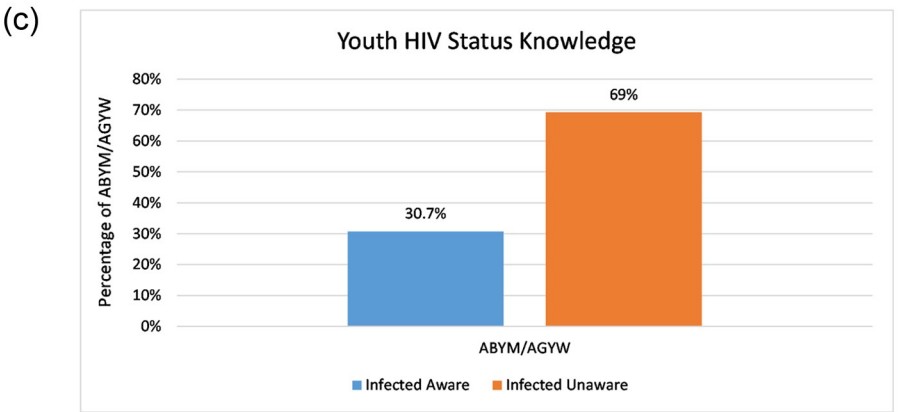

**Fig 1. Participant gender HIV status knowledge.** (a) Susceptible AGYW and ABYM HIV status knowledge, (b) Infected AGYW and ABYM HIV status knowledge, (c) AGYW/ABYM HIV status knowledge.

AGYW/ABYM UNAIDS-Kenya prevalence estimates and Fig 4(a) and 4(b) show the 1990—2018 UNAIDS-Kenya prevalence estimates for the Kenyan youth [47]. In South Africa, [38] fitted their mathematical model to UNAIDS HIV prevalence data to study the country's HIV epidemic trends. Hence, we used the 2012 KAIS data to inform the model formulation described in section 2.2 and some state variables initial conditions and, the UNAIDS-Kenya HIV Surveillance data to fit the model and estimate some of the model parameters.

(a)

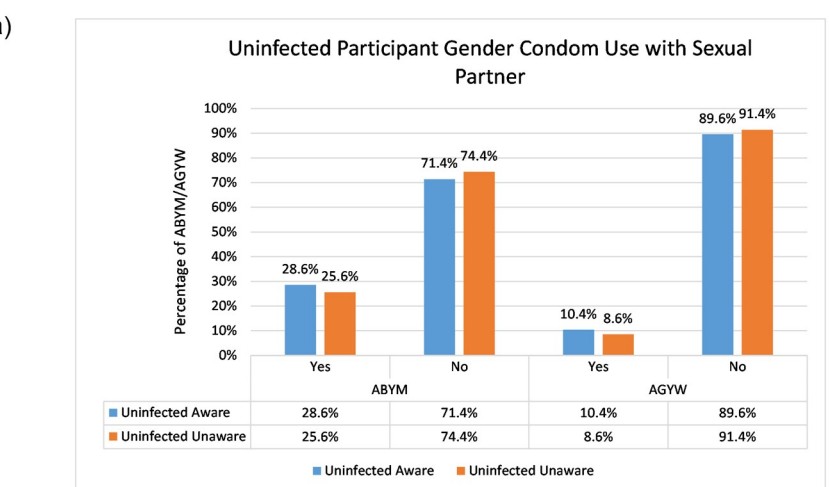

(b)

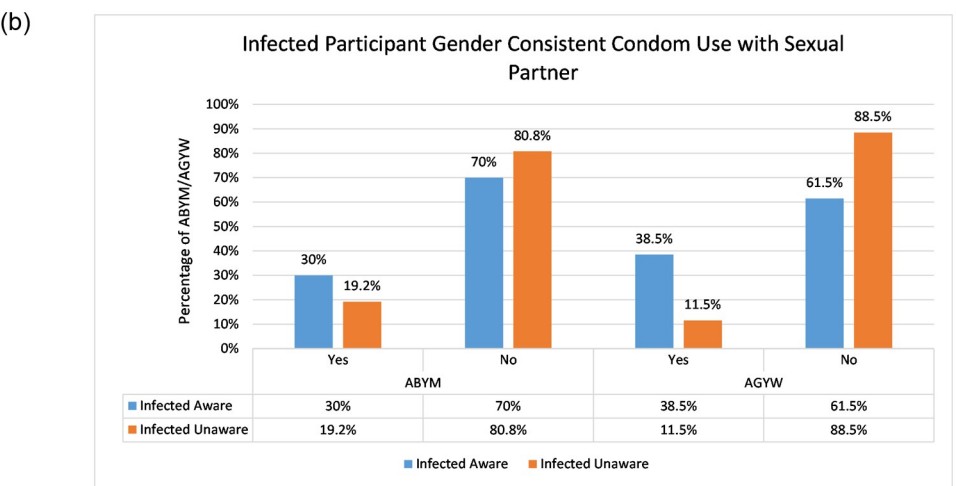

(c)

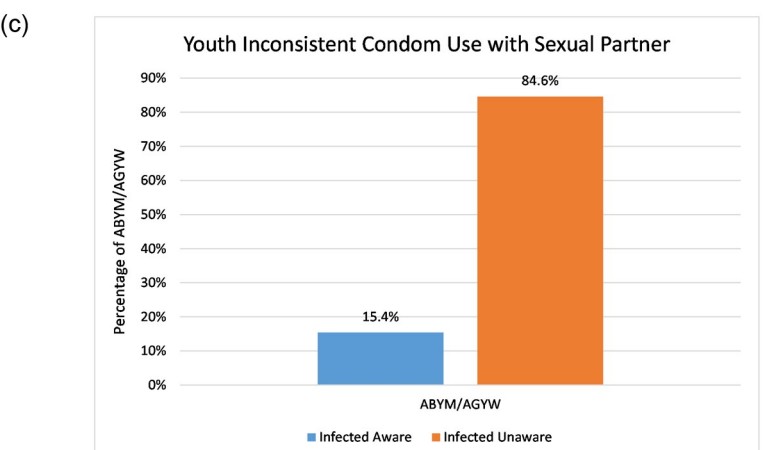

**Fig 2. Participant gender condom use patterns with sexual partner.** (a) Susceptible AGYW and ABYM condom use patterns with sexual partner, (b) Infected AGYW and ABYM condom use patterns with sexual partner, (c) Infected AGYW/ABYM inconsistent condom use patterns with sexual partner.

(a)

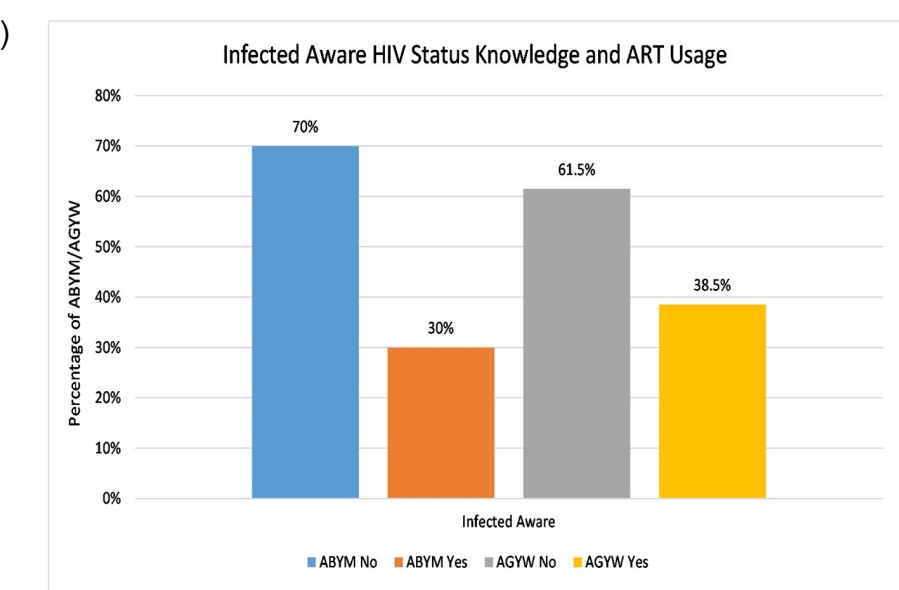

(b)

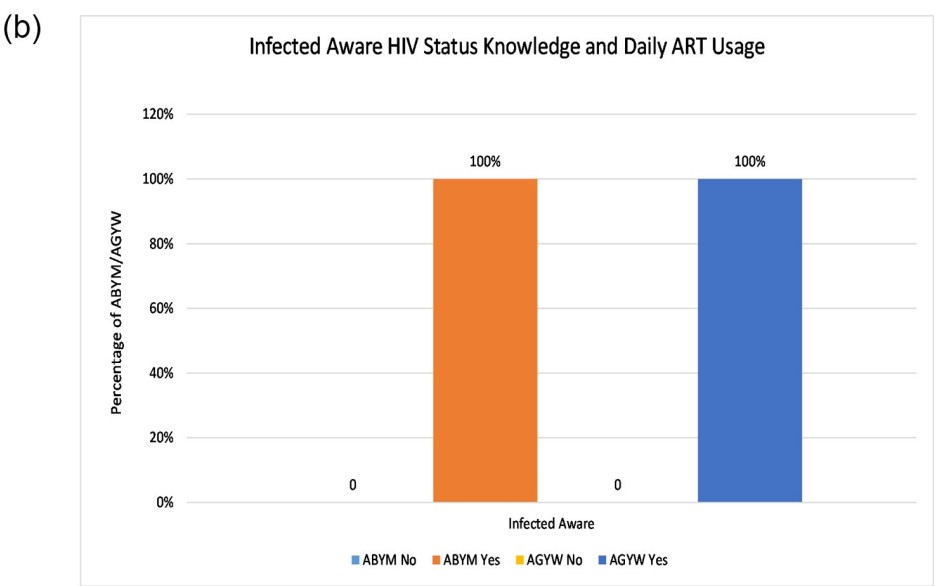

**Fig 3. HIV/AIDS infected participant gender ART usage.** (a) Infected AGYW and ABYM on ART, (b) Infected AGYW and ABYM daily ART usage.

## 2.2 Model formulation

We formulate a model describing HIV transmission dynamics in the AGYW and ABYM populations aged 15-24 with most of the state variables derived from the 2012 KAIS data described

**Table 1. 1990-2001 AGYW and ABYM UNAIDS-Kenya's prevalence data [47].**

| Year | 1990 | 1991 | 1992 | 1993 | 1994 | 1995 | 1996 | 1997 | 1998 | 1999 | 2000 | 2001 |
|---|---|---|---|---|---|---|---|---|---|---|---|---|
| AGYW Prevalence | 6.0 | 7.6 | 9.0 | 10.0 | 10.6 | 10.7 | 10.3 | 9.6 | 8.8 | 7.9 | 6.9 | 6.1 |
| ABYM Prevalence | 3.3 | 3.8 | 4.3 | 4.5 | 4.5 | 4.3 | 3.9 | 3.5 | 3.0 | 2.6 | 2.3 | 2.0 |

**Table 2. 2002-2013 AGYW and ABYM UNAIDS-Kenya's prevalence data [47].**

| Year | 2002 | 2003 | 2004 | 2005 | 2006 | 2007 | 2008 | 2009 | 2010 | 2011 | 2012 | 2013 |
|---|---|---|---|---|---|---|---|---|---|---|---|---|
| AGYW Prevalence | 5.4 | 4.8 | 4.3 | 4.0 | 3.7 | 3.5 | 3.4 | 3.3 | 3.2 | 3.1 | 3.0 | 3.0 |
| ABYM Prevalence | 1.7 | 1.6 | 1.4 | 1.4 | 1.4 | 1.4 | 1.4 | 1.5 | 1.5 | 1.5 | 1.6 | 1.6 |

**Table 3. 2014-2018 AGYW and ABYM UNAIDS-Kenya's prevalence data [47].**

| Year | 2014 | 2015 | 2016 | 2017 | 2018 |
|---|---|---|---|---|---|
| AGYW Prevalence | 2.9 | 2.8 | 2.7 | 2.6 | 2.5 |
| ABYM Prevalence | 1.6 | 1.6 | 1.6 | 1.6 | 1.5 |

in section 2.1.1 [44]. While all the infected aware on ART treatment remained adherent in section 2.1.1 and Fig 3, the model formulation considers the infected aware AGYW and ABYM populations on ART but are not adherent so as to make our model adaptable to non-adherence as the ART adherence rates among the infected aware youth in the KAIS data set was only for the 2012 data point. Section 1 highlights the need to model this population group as some of the infected aware youth on ART in general are not adherent to ART. Hence, we include this population group in the model formulation. We do not include the male population older than 24 years in this formulation as transactional sex in the 2012 KAIS population based survey was not common [48]. Hence, we primarily focus on the sexual behavior and use of HIV/AIDS control measures among the sexually active youth. In this study, the youth are defined as persons between the ages of 15 and 24 [49, 50].

The AGYW and ABYM populations are each categorized into six classes such that at time $t \geq 0$ there are susceptible AGYW, ABYM ($S_{fu}$, $S_{mu}$), infected AGYW, ABYM ($I_{fu}$, $I_{mu}$) who are not aware of their HIV status, susceptible AGYW, ABYM ($S_{fa}$, $S_{ma}$), infected AGYW, ABYM ($I_{fa}$, $I_{ma}$) who have tested for HIV/AIDS and are aware of their HIV status and use condoms consistently but are yet to be initiated on ART, infected AGYW, ABYM ($T_{fu}$, $T_{mu}$) who have tested for HIV/AIDS and are aware of their HIV status but use ART and condoms inconsistently and infected AGYW, ABYM ($T_{fa}$, $T_{ma}$) who have tested for HIV/AIDS and are aware of their HIV status and are adherent to ART and use condoms consistently. The total size of the AGYW and ABYM populations is given as $N_f = S_{fu} + S_{fa} + I_{fu} + I_{fa} + T_{fu} + T_{fa}$, $N_m = S_{mu} + S_{ma} + I_{mu} + I_{ma} + T_{mu} + T_{ma}$ respectively. $N = N_f + N_m$ is the total AGYW and ABYM population. Fig 5 represents the flow of individuals into different compartments in a single patch model.

The susceptibles females $S_{fu}$, $S_{fa}$, are free from the HIV infection but are at risk of infection through sexual contact with $I_{mu}$, $I_{ma}$ and $T_{mu}$ whereas the susceptibles males $S_{mu}$, $S_{ma}$, are free from the HIV infection but are at risk of infection through sexual contact with $I_{fu}$, $I_{fa}$ and $T_{fu}$. Infectivity in $I_{fu}$, $I_{mu}$ is much higher compared to $I_{fa}$, $I_{ma}$ and $T_{fu}$, $T_{mu}$ as the latter populations are more cautious given their infection status awareness compared to $I_{fu}$, $I_{mu}$ populations. Also, $T_{fu}$, $T_{mu}$ infectivity is further reduced given their partial use of condoms and ART compared to $I_{fa}$, $I_{ma}$ who partially use condoms for either pregnancy or HIV/AIDS protection. Perfect adherence of $T_{fa}$, $T_{ma}$ to condom use and ART reduces their viral load significantly such that they cannot sexually transmit HIV/AIDS given that undetectable viral load equals untransmittable [51]. Hence, we do not consider $T_{fa}$, $T_{ma}$ populations infectious in this model as their infectivity risks are negligible. The susceptible classes $S_{fu}$, $S_{mu}$ are at risk of infection at the incidence rates $\beta_{fu}$, $\beta_{mu}$, $\beta_{fa}$, $\beta_{ma}$ whereas $S_{fa}$, $S_{ma}$ are at risk of infection at the incidence

(a)

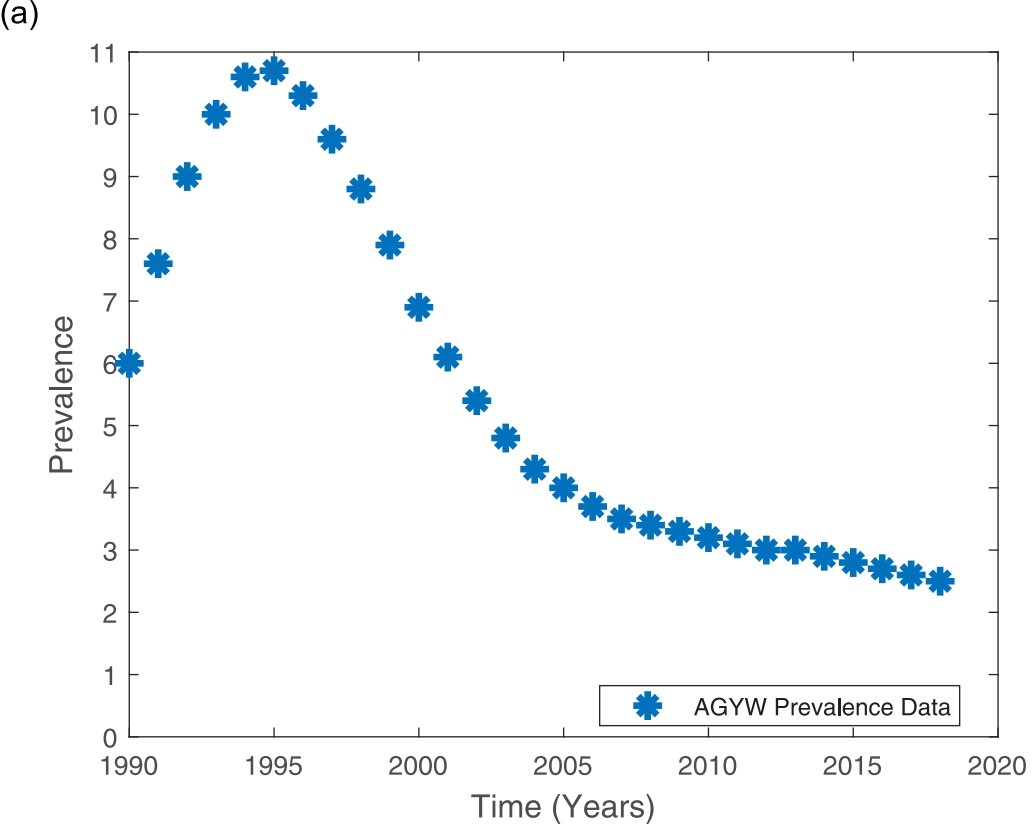

(b)

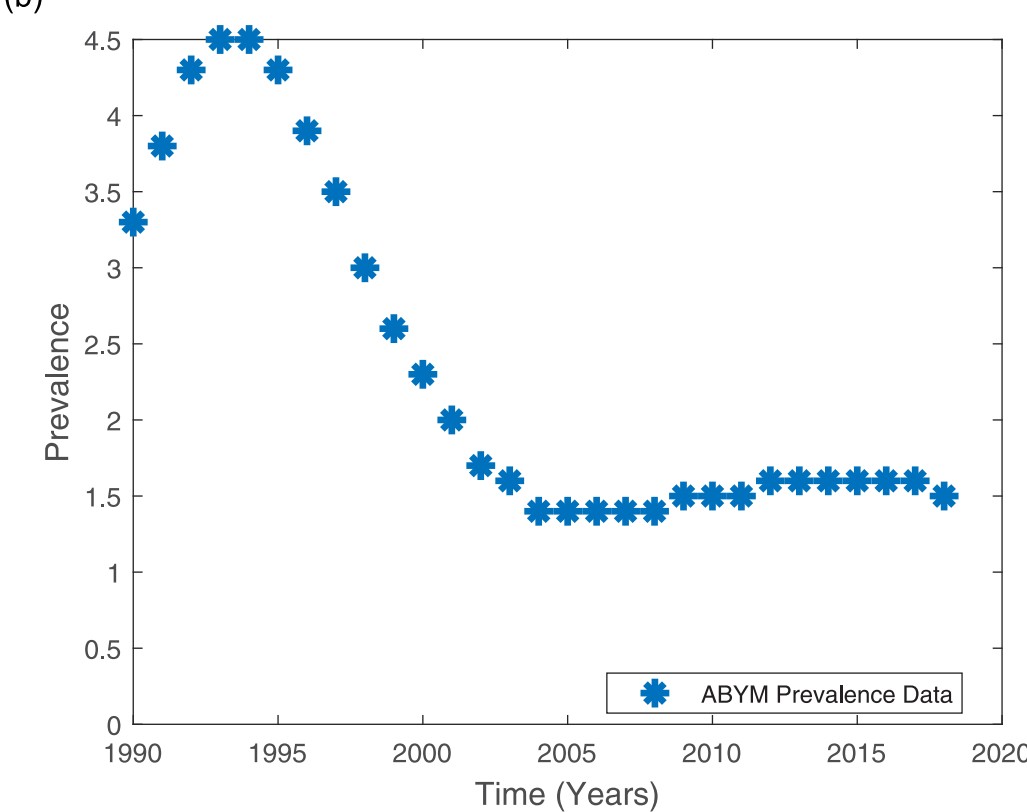

**Fig 4. AGYW and ABYM UNAIDS-Kenya 1990-2018 prevalence estimates** [47]. (a) AGYW UNAIDS-Kenya 1990—2018 Prevalence Estimates [47]. (b) ABYM UNAIDS-Kenya 1990—2018 Prevalence Estimates [47].

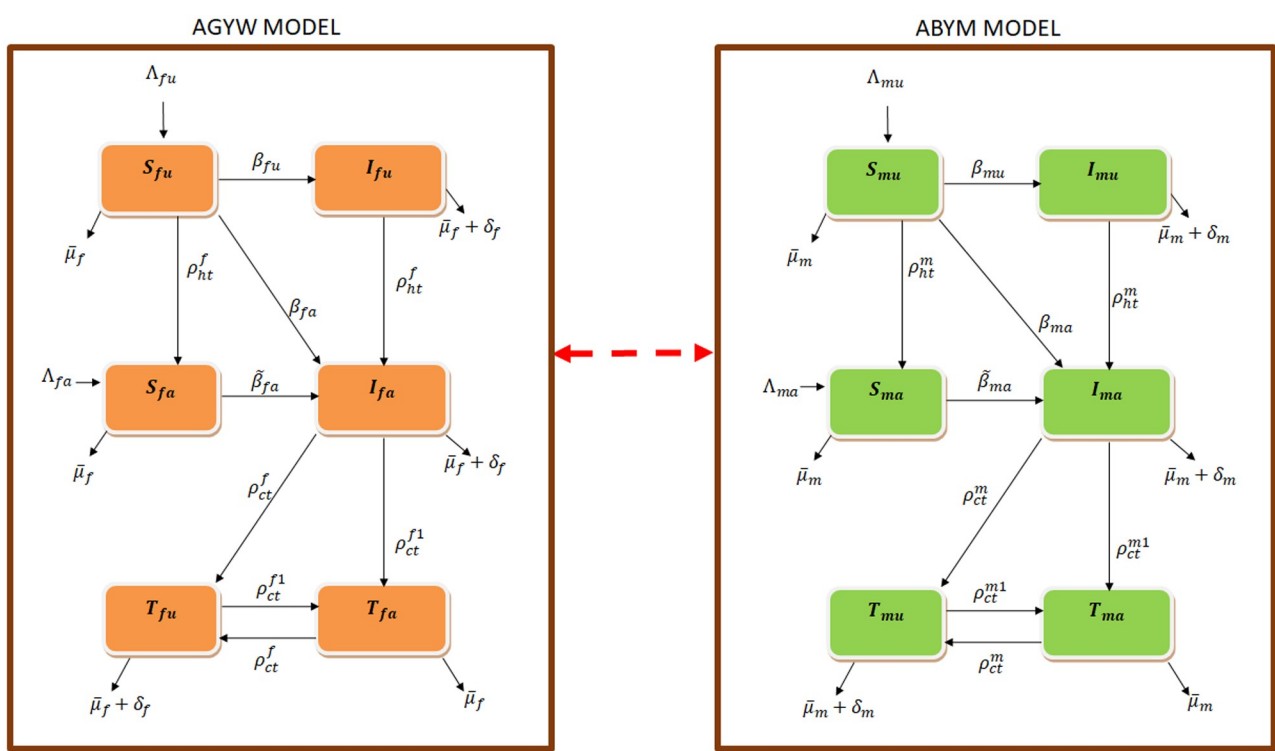

**Fig 5. Schematics of the compartmental model.** The AGYW and ABYM model describes the AGYW and ABYM transitions and interactions respectively.

rates $\tilde{\beta}_{fa}$, $\tilde{\beta}_{ma}$. The rates $\beta_{fu}$, $\beta_{mu}$, $\beta_{fa}$, $\beta_{ma}$, $\tilde{\beta}_{fa}$ and $\tilde{\beta}_{ma}$ are given in Eq (1) as

$$
\begin{cases}
\beta_{fu} = \dfrac{c_f \gamma_f}{N_m} \left[ I_{mu} + \alpha_c^m \rho_c I_{ma} + (\alpha_c^m \rho_c + \alpha_t^m \rho_t) T_{mu} \right], \\[2ex]
\beta_{fa} = \dfrac{c_f \gamma_f}{N_m} \left[ I_{mu} + \alpha_c^m \rho_c I_{ma} + (\alpha_c^m \rho_c + \alpha_t^m \rho_t) T_{mu} \right] \alpha_{ht}^m \rho_{ht}, \\[2ex]
\tilde{\beta}_{fa} = \dfrac{c_f \gamma_f}{N_m} \left[ I_{mu} + \alpha_c^m \rho_c I_{ma} + (\alpha_c^m \rho_c + \alpha_t^m \rho_t) T_{mu} \right] \alpha_{ht}^{m1} \rho_{ht}, \\[2ex]
\beta_{mu} = \dfrac{c_m \gamma_m}{N_f} \left[ I_{fu} + \alpha_c^f \rho_c I_{fa} + (\alpha_c^f \rho_c + \alpha_t^f \rho_t) T_{fu} \right], \\[2ex]
\beta_{ma} = \dfrac{c_m \gamma_m}{N_f} \left[ I_{fu} + \alpha_c^f \rho_c I_{fa} + (\alpha_c^f \rho_c + \alpha_t^f \rho_t) T_{fu} \right] \alpha_{ht}^f \rho_{ht}, \\[2ex]
\tilde{\beta}_{ma} = \dfrac{c_m \gamma_m}{N_f} \left[ I_{fu} + \alpha_c^f \rho_c I_{fa} + (\alpha_c^f \rho_c + \alpha_t^f \rho_t) T_{fu} \right] \alpha_{ht}^{f1} \rho_{ht}.
\end{cases}
\tag{1}
$$

Contacts $c_f$, $c_m$ are the average number of sexual interactions by AGYW/ABYM with individuals of the opposite sex per unit time whereas $\gamma_f$, $\gamma_m$ are the probabilities that a susceptible AGYW/ABYM coming into proper contact with an infected individual of the opposite sex per unit time will contract the disease. Condom use rate ($\rho_c$) decreases the disease spread by $I_{fa}$, $I_{ma}$ whereas condom use and ART adherence rate ($\rho_t$) reduces the infection risk by $T_{fu}$, $T_{mu}$.

HIV/AIDS status disclosure ($\rho_{ht}$) by newly HIV/AIDS tested $I_{fu}$, $I_{mu}$ and already tested populations $I_{fa}$, $I_{ma}$, $T_{fu}$, $T_{mu}$ further reduces the disease spread to the susceptible populations.

When each of the HIV/AIDS control measures $\rho_{ht}$, $\rho_c$, $\rho_t$ in the AGYW/ABYM populations is 1 we have perfect adherence otherwise, $0 \leq \rho_{ht}, \rho_c, \rho_t < 1$. The rates $\alpha_{ht}^f$, $\alpha_{ht}^m$ represent negative attitudes affecting the efficacy of HIV testing rate $\rho_{ht}$ in the AGYW and ABYM populations such as poor health services, poverty, psycho-social conditions, socio-demographic characteristics among others [8–10]. Rates $\alpha_c^f$, $\alpha_c^m$ represent negative attitudes affecting the efficacy of condom use rate in the AGYW and ABYM populations such as religion, peer influence, perceived individual's risk among others [3, 12–15]. Also, $\alpha_t^f$, $\alpha_t^m$ represent negative attitudes affecting the efficacy of ART usage rate among the infected AGYW and ABYM such as stigma, poverty, caregivers waning support, confidentiality breaches by health workers among others [9, 10, 20, 21]. Section 1 highlights how societal attitudes affect HIV testing rates, condom use and adherence to ART among the youth in Kenya. The rates $\alpha_c^f \rho_c$, $\alpha_c^m \rho_c$ acts on $I_{fa}$, $I_{ma}$ to reduce their infectivity as condom use serves to protect susceptible AGYW and ABYM from acquiring new HIV/AIDS infection. In addition to condom use, $T_{fu}$, $T_{mu}$ partially uses ART which works to reduce their HIV/AIDS viral load. The combined effects of condom use and ART usage ($\alpha_c^f \rho_c + \alpha_t^f \rho_t$, $\alpha_c^m \rho_c + \alpha_t^m \rho_t$) further reduces the infectivity of $T_{fu}$, $T_{mu}$ as $0 < \alpha_c^f, \alpha_c^m, \alpha_t^f, \alpha_t^m < 1$. Thus, $T_{fu}$, $T_{mu}$ infectivity is less than $I_{fa}$, $I_{ma}$ which is less than $I_{fu}$, $I_{mu}$.

Incidence rates by untested AGYW/ABYM with individuals of the opposite sex per unit time are given as $\beta_{fu}$, $\beta_{mu}$ respectively. The incidence rates $\beta_{fa}$, $\beta_{ma}$ are given by HIV/AIDS tested AGYW/ABYM but not under ART treatment with individuals of the opposite sex per unit time. The incidence rates $\tilde{\beta}_{fa}$, $\tilde{\beta}_{ma}$ results from HIV/AIDS tested youth who are not perfectly adherent to consistent condom use and ART treatment with individuals of the opposite sex per unit time. The incidence rates $\beta_{fu}$, $\beta_{mu}$, $\beta_{fa}$, $\beta_{ma}$, $\tilde{\beta}_{fa}$ and $\tilde{\beta}_{ma}$ have proportionate mixing incidences since some of the youth aged 15-24 will have already initiated sex with most of them remaining sexually active.

Uninfected unaware $S_{fu}$, $S_{mu}$ who know their HIV/AIDS status through HIV testing moves to $S_{fa}$, $S_{ma}$ at the rates $\rho_{ht}^f$, $\rho_{ht}^m$ with $\rho_{ht}^f = \alpha_{ht}^f \rho_{ht}$ and $\rho_{ht}^m = \alpha_{ht}^m \rho_{ht}$. A newly infected $S_{fu}$, $S_{mu}$ through interaction with infected $I_{mu}$, $I_{ma}$ or $T_{mu}$ who fail to disclose their HIV/AIDS status will move to $I_{fu}$, $I_{mu}$ at the rates $\beta_{fu}$, $\beta_{mu}$. Also, a newly infected $S_{fu}$, $S_{mu}$ through sexual contact with infected aware populations of the opposite sex will move to $I_{fa}$, $I_{ma}$ at the rates $\beta_{fa}$, $\beta_{ma}$ given that status disclosure by the infected aware populations results in HIV/AIDS awareness of the newly infected $S_{fu}$, $S_{mu}$. A newly infected $S_{fa}$, $S_{ma}$ moves to $I_{fa}$, $I_{ma}$ at the rates $\tilde{\beta}_{fa}$, $\tilde{\beta}_{ma}$. Infected unaware $I_{fu}$, $I_{mu}$ can move to $I_{fa}$, $I_{ma}$ at the rates $\rho_{ht}^f$, $\rho_{ht}^m$ through HIV testing. Also, $I_{fa}$, $I_{ma}$ and $T_{fu}$, $T_{mu}$ who consistently use condoms and adhere to ART treatment moves to $T_{fa}$, $T_{ma}$ at the rates $\rho_{ct}^{f1}$, $\rho_{ct}^{m1}$ whereas an $I_{fa}$, $I_{ma}$ or $T_{fa}$, $T_{ma}$ who fail to use condoms consistently or adhere to ART treatment moves to $T_{fu}$, $T_{mu}$ at the rates $\rho_{ct}^f$, $\rho_{ct}^m$ respectively with $\rho_{ct}^{f1} = \alpha_c^{f1} \rho_c + \alpha_t^{f1} \rho_t$, $\rho_{ct}^{m1} = \alpha_c^{m1} \rho_c + \alpha_t^{m1} \rho_t$, $\rho_{ct}^f = \alpha_c^f \rho_c + \alpha_t^f \rho_t$ and $\rho_{ct}^m = \alpha_c^m \rho_c + \alpha_t^m \rho_t$ respectively. $\alpha_{ht}^{f1}$, $\alpha_{ht}^{m1}$, $\alpha_c^{f1}$, $\alpha_t^{f1}$, $\alpha_c^{m1}$, $\alpha_t^{m1}$ and $\alpha_{ht}^f$, $\alpha_{ht}^m$, $\alpha_c^f$, $\alpha_t^f$, $\alpha_c^m$, $\alpha_t^m$ are parameters representing negative/positive attitudes influencing HIV/AIDS control measures ($\rho_{ht}$, $\rho_c$, $\rho_t$) but not to zero given that in the Kenyan HIV/AIDS youth dynamics some control measures are in place [45]. The rates $\alpha_{ht}^{f1}$, $\alpha_{ht}^{m1}$, $\alpha_c^{f1}$, $\alpha_t^{f1}$, $\alpha_c^{m1}$ represent attitudes affecting the efficacy of $\rho_{ht}$, $\rho_c$, $\rho_t$ positively such as confidentiality by health workers, adequate support structure at home and the community at large, improved financial status among others whereas $\alpha_{ht}^f$, $\alpha_{ht}^m$, $\alpha_c^f$, $\alpha_t^f$, $\alpha_c^m$ represent negative attitudes, which was explained earlier, influencing the said control measures. The rates $\rho_{ct}^f$, $\rho_{ct}^m$ represent combined condom use and ART use coupled with negative attitudes whereas

$\rho_{ct}^{f1}$, $\rho_{ct}^{m1}$ represent combined condom use and ART use coupled with positive attitudes among the AGYW and ABYM respectively. Thus,

$$0 < \alpha_{ht}^{f1}, \ \alpha_{ht}^{m1}, \ \alpha_{c}^{f1}, \ \alpha_{t}^{f1}, \ \alpha_{c}^{m1}, \ \alpha_{t}^{m1}, \ \alpha_{ht}^{f}, \ \alpha_{ht}^{m}, \ \alpha_{c}^{f}, \ \alpha_{t}^{f}, \ \alpha_{c}^{m}, \ \alpha_{t}^{m} < 1$$

with

$$\alpha_{ht}^{f1}, \ \alpha_{ht}^{m1}, \ \alpha_{c}^{f1}, \ \alpha_{t}^{f1}, \ \alpha_{c}^{m1}, \ \alpha_{t}^{m1} \ > \ \alpha_{ht}^{f}, \ \alpha_{ht}^{m}, \ \alpha_{c}^{f}, \ \alpha_{t}^{f}, \ \alpha_{c}^{m}, \ \alpha_{t}^{m}.$$

Recruitment rates into susceptible populations $S_{fu}$, $S_{mu}$, $S_{fa}$, $S_{ma}$ is by natural births and maturity to 15 years and are given as $\Lambda_{fu}$, $\Lambda_{mu}$, $\Lambda_{fa}$, $\Lambda_{ma}$ respectively. The susceptible classes are all reduced by natural deaths $\mu_f$, $\mu_m$ whereas the infectious classes are all decreased by natural deaths and disease induced deaths, $\delta_f$, $\delta_m$. Upon turning 24 years, the AGYW and the ABYM population exit the model at the rate $\sigma$. The state variables and parameters are assumed to be positive given that a population dynamics model is being studied. Tables 4 and 5 gives the summary description for the state variables and model parameters respectively.

The system of ordinary differential equations governing the AGYW/ABYM HIV model is given by the system of Eq (2) as

$$
\begin{cases}
\dfrac{dS_{fu}}{dt} = \Lambda_{fu} - \beta_{fu}\, S_{fu} - \beta_{fa}\, S_{fu} - \mu_{f1}\, S_{fu}, \\[2mm]
\dfrac{dS_{fa}}{dt} = \Lambda_{fa} + \rho_{ht}^{f}\, S_{fu} - \tilde{\beta}_{fa}\, S_{fa} - \mu_{f2}\, S_{fa}, \\[2mm]
\dfrac{dI_{fu}}{dt} = \beta_{fu}\, S_{fu} - \mu_{f3}\, I_{fu}, \\[2mm]
\dfrac{dI_{fa}}{dt} = \tilde{\beta}_{fa}\, S_{fa} + \beta_{fa}\, S_{fu} + \rho_{ht}^{f}\, I_{fu} - \mu_{f4}\, I_{fa}, \\[2mm]
\dfrac{dT_{fu}}{dt} = \rho_{ct}^{f}\, I_{fa} + \rho_{ct}^{f}\, T_{fa} - \mu_{f5}\, T_{fu}, \\[2mm]
\dfrac{dT_{fa}}{dt} = \rho_{ct}^{f1} I_{fa} + \rho_{ct}^{f1}\, T_{fu} - \mu_{f6}\, T_{fa}, \\[2mm]
\dfrac{dS_{mu}}{dt} = \Lambda_{mu} - \beta_{mu}\, S_{mu} - \beta_{ma}\, S_{mu} - \mu_{m1}\, S_{mu}, \\[2mm]
\dfrac{dS_{ma}}{dt} = \Lambda_{ma} + \rho_{ht}^{m}\, S_{mu} - \tilde{\beta}_{ma}\, S_{ma} - \mu_{m2}\, S_{ma}, \\[2mm]
\dfrac{dI_{mu}}{dt} = \beta_{mu}\, S_{mu} - \mu_{m3}\, I_{mu}, \\[2mm]
\dfrac{dI_{ma}}{dt} = \tilde{\beta}_{ma}\, S_{ma} + \beta_{ma}\, S_{mu} + \rho_{ht}^{m}\, I_{mu} - \mu_{m4}\, I_{ma}, \\[2mm]
\dfrac{dT_{mu}}{dt} = \rho_{ct}^{m}\, I_{ma} + \rho_{ct}^{m}\, T_{ma} - \mu_{m5}\, T_{mu}, \\[2mm]
\dfrac{dT_{ma}}{dt} = \rho_{ct}^{m1}\, I_{ma} + \rho_{ct}^{m1}\, T_{mu} - \mu_{m6}\, T_{ma}.
\end{cases}
\tag{2}
$$

where $\bar{\mu}_f = \mu_f + \sigma$, $\mu_{f1} = \rho_{ht}^{f} + \bar{\mu}_f$, $\mu_{f2} = \bar{\mu}_f$, $\mu_{f3} = \rho_{ht}^{f} + \bar{\mu}_f + \delta_f$, $\mu_{f4} = \rho_{ct}^{f} + \rho_{ct}^{f1} + \bar{\mu}_f + \delta_f$, $\mu_{f5} = \rho_{ct}^{f1} + \bar{\mu}_f + \delta_f$, $\mu_{f6} = \rho_{ct}^{f} + \bar{\mu}_f$, $\bar{\mu}_m = \mu_m + \sigma$, $\mu_{m1} = \rho_{ht}^{m} + \bar{\mu}_m$, $\mu_{m2} = \bar{\mu}_m$, $\mu_{m3} = \rho_{ht}^{m} + \bar{\mu}_m + \delta_m$, $\mu_{m4} = \rho_{ct}^{m} + \rho_{ct}^{m1} + \bar{\mu}_m + \delta_m$, $\mu_{m5} = \rho_{ct}^{m1} + \bar{\mu}_m + \delta_m$, $\mu_{m6} = \rho_{ct}^{m} + \bar{\mu}_m$.

**Table 4. Summary description of state variables.**

| Variable | Description |
|---|---|
| $S_{fu}$, $S_{mu}$ | Susceptible AGYW & ABYM who have never tested for HIV/AIDS |
| $S_{fa}$, $S_{ma}$ | Susceptible AGYW & ABYM who have ever tested for HIV/AIDS |
| $I_{fu}$, $I_{mu}$ | Infected AGYW & ABYM who have never tested for HIV/AIDS |
| $I_{fa}$, $I_{ma}$ | Infected AGYW & ABYM who have ever tested for HIV/AIDS |
| $T_{fu}$, $T_{mu}$ | Infected aware AGYW & ABYM who are not adherent to ART or consistent condom use |
| $T_{fa}$, $T_{ma}$ | Infected aware AGYW & ABYM who are adherent to ART and use condoms consistently |

**Table 5. Summary description of parameters.**

| Parameter | Description |
|---|---|
| $\Lambda_{fu}$, $\Lambda_{mu}$ | Natural birth and maturity rates of susceptible AGYW and ABYM unaware of their HIV status |
| $\Lambda_{fa}$, $\Lambda_{ma}$ | Natural birth and maturity rates of susceptible AGYW and ABYM aware of their HIV status |
| $\rho_{ht}$ | AGYW/ABYM HIV testing rate |
| $\rho_t$ | AGYW/ABYM adherence rate to anti-retroviral therapy treatment |
| $\rho_c$ | AGYW/ABYM condom use rate |
| $\mu_f$, $\mu_m$ | Natural death rates of AGYW and ABYM respectively |
| $\gamma_f$, $\gamma_m$ | Probabilities of AGYW and ABYM transmission risk |
| $\delta_f$, $\delta_m$ | Disease induced deaths in AGYW and ABYM respectively |
| $c_f$, $c_m$ | AGYW and ABYM sexual contact rates |
| $\alpha_{ht}^f$, $\alpha_{ht}^m$, $\alpha_{ht}^{f1}$, $\alpha_{ht}^{m1}$ | Negative and positive attitude rates influencing HIV testing rates among the AGYW and ABYM respectively |
| $\alpha_c^f$, $\alpha_c^m$, $\alpha_c^{f1}$, $\alpha_c^{m1}$ | Negative and positive attitude rates influencing condom use rates among the AGYW and ABYM respectively |
| $\alpha_t^f$, $\alpha_t^m$, $\alpha_t^{f1}$, $\alpha_t^{m1}$ | Negative and positive attitude rates influencing ART adherence rates among the AGYW and ABYM respectively |
| $\sigma$ | Exit rate of AGYW and ABYM upon turning 24 years |

## 2.3 Model properties

Mathematical analysis of the formulated model system (2) is presented here. We show that the compact system of ordinary differential Eq (2) governing the model of biological interest is well-posed and control reproduction number with its biological interpretation given. The conditions for stability of the model steady states are determined.

**2.3.1 Boundedness.** **Theorem 2.1** *The model* (2) *solutions are uniformly bounded in a set*

$$\Omega = \left\{ (S_{fu}, S_{fa}, I_{fu}, I_{fa}, T_{fu}, T_{fa}, S_{mu}, S_{ma}, I_{mu}, I_{ma}, T_{mu}, T_{ma}) \in \mathbb{R}_{12}^+ \mid N(0) \le N \le \frac{\tilde{\Lambda}}{\mu_f + \mu_m} \right\}.$$

**Proof 2.1** *Given that system* (2) *is a finite dimensional dynamical system, its initial conditions and boundary conditions need to be constrained to* $\Omega$. *Let* $(S_{fu}, S_{fa}, I_{fu}, I_{fa}, T_{fu}, T_{fa}, S_{mu}, S_{ma}, I_{mu},$ $I_{ma}, T_{mu}, T_{ma})$ *be the solution to* (2) *and* $S_{fu}(0) = S_{fu}^0 \ge 0$, $S_{fa}(0) = S_{fa}^0 \ge 0$, $I_{fu}(0) = I_{fu}^0 \ge$ $0$, $I_{fa}(0) = I_{fa}^0 \ge 0$, $T_{fu}(0) = T_{fu}^0 \ge 0$, $T_{fa}(0) = T_{fa}^0 \ge 0$ $S_{mu}(0) = S_{mu}^0 \ge 0$, $S_{ma}(0) = S_{ma}^0 \ge$ $0$, $I_{mu}(0) = I_{mu}^0 \ge 0$, $I_{ma}(0) = I_{ma}^0 \ge 0$, $T_{mu}(0) = T_{mu}^0 \ge 0$, $T_{ma}(0) = T_{ma}^0 \ge 0$ *be the initial conditions. Adding all equations of system* (2), *yields*

$$\dot{N} = (\tilde{\Lambda}) - \bar{\mu}_f N_f - \bar{\mu}_m N_m - \delta_f (N_f - \tilde{N}_f) - \delta_m (N_m - \tilde{N}_m)$$

$$\le \tilde{\Lambda} - (\bar{\mu}_f + \delta_f)N_f - (\bar{\mu}_m + \delta_m)N_m - \delta_f \tilde{N}_f - \delta_m \tilde{N}_m$$

$$\le \tilde{\Lambda} - \tilde{\mu} N$$

*where* $\tilde{\Lambda} = \Lambda_{fu} + \Lambda_{fa} + \Lambda_{mu} + \Lambda_{ma}$, $\tilde{N}_f = S_{fu} + S_{fa} + T_{fa}$, $\tilde{N}_m = S_{mu} + S_{ma} + T_{ma}$,

$\tilde{\mu} = min(\bar{\mu}_f + \delta_f, \bar{\mu}_m + \delta_m)$. *Thus, $\Omega$ is a compact attracting non-negatively invariant for positive starting-point values since $N(0)>0$. This can easily be proved using the theory of differential inequality* [52]. *All solutions of* (2) *originating in $\mathbb{R}_+^{12}$ are confined in $\Omega$. Let M be an upper bound for $S_{fu}, S_{fa}, I_{fu}, I_{fa}, T_{fu}, T_{fa}, S_{mu}, S_{ma}, I_{mu}, I_{ma}, T_{mu}, T_{ma}$. We then conclude that every solution originating from $\Omega$ stays in $\Omega$ and is bounded by M.*

**2.3.2 Local existence and uniqueness.** **Lemma 2.1** *Let $x = (x_i)_{i = 1,2,\ldots,12}$ and $f$ :*
$\mathbb{R}_+ \times \mathbb{R}^{12} \to \mathbb{R}^{12}$ *be continuous with respect to t, x and Lipschitz continuous. Let $f(t, x)$ be non negative for all $(t, x) \in \mathbb{R}_+ \times \mathbb{R}^{12}$, and $x_i = 0$. For every $x_0 \in \mathbb{R}_+^{12}$, there exists a positive constant T such that $\dot{x} = f(t, x)$, $x(t_0) = x_0$, has a unique, positive and existing solution whose value lies in the interval $[0, T)$ and in $\mathbb{R}_+^{12}$. If $T < \infty$ then $\lim \sup_{t \to T} \sum_{i=1}^{12} x_i = +\infty$.*

**Theorem 2.2** *The solution set $\{S_{fu}, S_{fa}, I_{fu}, I_{fa}, T_{fu}, T_{fa}, S_{mu}, S_{ma}, I_{mu}, I_{ma}, T_{mu}, T_{ma}\}$ of the model* (2) *exists, is unique and positive for $t > 0$.*

By theorem 2.1, the solutions to (2) are uniformly bounded on $[0, T]$. By theorem 2.2, the solution of (2) exists for any finite time. Thus, for any positive initial data in $\mathbb{R}_+^{12}$, the model system (2) will possess a unique and positive solution in $\mathbb{R}_+^{12}$. This proves that all feasible solution of the model system (2) lies in the feasible region, $\Omega$.

**2.3.3 Equilibria.** The model system (2) has a unique disease-free equilibrium (DFE)

$$E^0 = (S_{fu}^0, S_{fa}^0, 0, 0, 0, 0, S_{mu}^0, S_{ma}^0, 0, 0, 0, 0)$$

and possibly an endemic equilibrium (EE)

$$E^* = (S_{fu}^*, S_{fa}^*, I_{fu}^*, I_{fa}^*, T_{fu}^*, T_{fa}^*, S_{mu}^*, S_{ma}^*, I_{mu}^*, I_{ma}^*, T_{mu}^*, T_{ma}^*)$$

with

$$
\begin{cases}
S_{fu}^0 = \dfrac{\Lambda_{fu}}{\mu_{f1}}, \quad S_{fa}^0 = \dfrac{\Lambda_{fa}\,\mu_{f1} + \rho_{ht}^f \Lambda_{fu}}{\mu_{f1}\,\mu_{f2}}, \\[2ex]
S_{mu}^0 = \dfrac{\Lambda_{mu}}{\mu_{m1}}, \quad S_{ma}^0 = \dfrac{\Lambda_{ma}\,\mu_{m1} + \rho_{ht}^m \Lambda_{mu}}{\mu_{m1}\,\mu_{m2}}, \\[2ex]
S_{fu}^* = \dfrac{\Lambda_{fu}}{g_{02}\beta_{fu}^* + \mu_{f1}}, \quad S_{fa}^* = \dfrac{\Lambda_{fa}}{\rho_{ht}^{m1}\beta_{fu}* + \mu_{f2}} + \dfrac{\rho_{ht}^f \Lambda_{fu}}{(\rho_{ht}^{m1}\beta_{fu}^* + \mu_{f2})(g_{02}\beta_{fu}^* + \mu_{f1})}, \\[2ex]
I_{fu}^* = \dfrac{\Lambda_{fu}\beta_{fu}^*}{\mu_{f3}\,(g_{02}\beta_{fu}^* + \mu_{f1})}, \quad I_{fa}^* = \dfrac{q_{02}\beta_{fu}^{*2} + q_{03}\beta_{fu}^* + q_{04}}{q_{05}\beta_{fu}^{*2} + q_{06}\beta_{fu}^* + q_{07}}, \quad T_{fu}^* = g_{01} I_{fa}^*, \quad T_{fa}^* = g_{00} I_{fa}^*, \\[2ex]
S_{mu}^* = \dfrac{\Lambda_{mu}}{g_{08}\beta_{mu}^* + \mu_{m1}}, \quad S_{ma}^* = \dfrac{\Lambda_{ma}}{\rho_{ht}^{f1}\beta_{mu}* + \mu_{m2}} + \dfrac{\rho_{ht}^m \Lambda_{mu}}{(\rho_{ht}^{f1}\beta_{mu}^* + \mu_{m2})(g_{08}\beta_{mu}^* + \mu_{m1})}, \quad (3) \\[2ex]
I_{mu}^* = \dfrac{\Lambda_{mu}\beta_{mu}^*}{\mu_{m3}\,(g_{08}\beta_{mu}^* + \mu_{m1})}, \quad I_{ma}^* = \dfrac{h_{02}\beta_{mu}^{*2} + h_{03}\beta_{mu}^* + h_{04}}{h_{05}\beta_{mu}^{*2} + h_{06}\beta_{mu}^* + h_{07}}, \quad T_{mu}^* = g_{07} I_{ma}^*, \\[2ex]
T_{ma}^* = g_{06} I_{ma}^*, \quad N_f^* = \dfrac{\Lambda_{fu} + \Lambda_{fa} + \delta_f \tilde{N}_f^*}{\bar{\mu}_f + \delta_f}, \quad \tilde{N}_f^* = S_{fu}^* + S_{fa}^* + T_{fa}^*, \\[2ex]
N_m^* = \dfrac{\Lambda_{mu} + \Lambda_{ma} + \delta_m \tilde{N}_m^*}{\bar{\mu}_m + \delta_m}, \quad \tilde{N}_m^* = S_{mu}^* + S_{ma}^* + T_{ma}^*, \\[2ex]
\beta_{fu}^{*5} + C_1 \beta_{fu}^{*4} + C_2 \beta_{fu}^{*3} + C_3 \beta_{fu}^{*2} + C_4 \beta_{fu}^* - C_5 = 0, \\[2ex]
\beta_{mu}^{*5} + C_{11} \beta_{mu}^{*4} + C_{21} \beta_{mu}^{*3} + C_{31} \beta_{mu}^{*2} + C_{41} \beta_{mu}^* - C_{51} = 0.
\end{cases}
$$

Refer to S1 Appendix for the expressions of $g_{00}, g_{01}, \ldots, g_{11}, q_{01}, q_{02}, \ldots, q_{20}, h_{01}, h_{02}, \ldots,$ $h_{20}, C_1, C_2, \ldots, C_5$ and $C_{11}, C_{21}, \ldots, C_{51}$.

By the fundamental theorem of algebra, the polynomial equations $\beta_{fu}^{*5} + C_1 \beta_{fu}^{*4} + C_2 \beta_{fu}^{*3} + C_3 \beta_{fu}^{*2} + C_4 \beta_{fu}^{*} - C_5 = 0$ and $\beta_{mu}^{*5} + C_{11} \beta_{mu}^{*4} + C_{21} \beta_{mu}^{*3} + C_{31} \beta_{mu}^{*2} + C_{41} \beta_{mu}^{*} - C_{51} = 0$, of odd degree, have at least one real root each. By Descartes' rule of signs, the polynomial equations will each have at least one non-negative real root if and only if $C_1 > 0, C_2 > 0, C_3 > 0, C_4 > 0,$ $C_5 > 0$ and $C_{11} > 0, C_{21} > 0, C_{31} > 0, C_{41} > 0, C_{51} > 0$, given that the sign before $C_5$ and $C_{51}$ is negative and the sign before $\beta_{fu}^{*5}$ and $\beta_{mu}^{*5}$ is non-negative otherwise each of the polynomial equation will have at most four (4) non-negative real roots. The exact number of non-negative roots can be determined using Descartes' rule of signs and Euclid's algorithm of the Sturm's theorem.

## 2.4 Control reproduction number, $\mathcal{R}_c$

The control reproduction number, $\mathcal{R}_c$, is defined as the expected number of secondary infections produced by a typical infected individual during its entire period of infectiousness in a population that is not entirely susceptible due to the presence of control measures 53. The control measures present in our model are HIV testing ($\rho_{ht}$), condom use ($\rho_c$) and ART adherence ($\rho_t$).

The global dynamics for many disease models is determined by the sharp threshold criterion given by the basic reproduction number and this is true for our model system (2) [54]. Model system (2) possesses a sharp threshold if the control reproduction number $\mathcal{R}_c$ given by Eq 7 is such that $E^0$ is globally attractive for $\mathcal{R}_c \leq 1$ and there is a unique endemic equilibrium $E^*$ that is globally attractive in the feasible region for $\mathcal{R}_c > 1$. Biologically, $\mathcal{R}_c$ is used to measure the transmission potential of the HIV/AIDS disease among the AGYW and ABYM in the presence of the said control measures [54]. The threshold property states that if $\mathcal{R}_c > 1$, HIV/AIDS disease persists in the youthful population hence becoming endemic whereas when $\mathcal{R}_c < 1$, the disease mirrors the effects of successful combination control measures to the AGYW and ABYM consequently protecting the susceptible youth from acquiring new HIV/AIDS infection.

The next generation matrix approach is used to compute the control reproduction number for the model system (2) [54]. Consider the infected subsystem of the model system (2) given as

$$
\begin{cases}
\dfrac{dI_{fu}}{dt} = \beta_{fu} S_{fu} - \mu_{f3} I_{fu}, \\[2mm]
\dfrac{dI_{fa}}{dt} = \tilde{\beta}_{fa} S_{fa} + \beta_{fa} S_{fu} + \rho_{ht}^f I_{fu} - \mu_{f4} I_{fa}, \\[2mm]
\dfrac{dT_{fu}}{dt} = \rho_{ct}^f I_{fa} + \rho_{ct}^f T_{fa} - \mu_{f5} T_{fu}, \\[2mm]
\dfrac{dI_{mu}}{dt} = \beta_{mu} S_{mu} - \mu_{m3} I_{mu}, \\[2mm]
\dfrac{dI_{ma}}{dt} = \tilde{\beta}_{ma} S_{ma} + \beta_{ma} S_{mu} + \rho_{ht}^m I_{mu} - \mu_{m4} I_{ma}, \\[2mm]
\dfrac{dT_{mu}}{dt} = \rho_{ct}^m I_{ma} + \rho_{ct}^m T_{ma} - \mu_{m5} T_{mu}.
\end{cases}
\tag{4}
$$

The right hand side of the infected subsystem (4) is decomposed into two parts, $F$ and $V$ where $F$ denotes the transmission part and each $F_i$ represents new infection. $V$ denotes the

transition part and each $V_i$ describes change in state for instance removal through natural deaths, disease induced deaths, aging, HIV/AIDS status knowledge, condom use and ART adherence [55].

$$
F = \begin{bmatrix}
\left( \dfrac{c_f \gamma_f}{N_m} \left[ I_{mu} + \alpha_c^m \rho_c^m I_{ma} + (\alpha_c^m \rho_c^m + \alpha_t^m \rho_t^m) T_{mu} \right] \right) S_{fu} \\[2ex]
\rho_{ht}^m \left( \dfrac{c_f \gamma_f}{N_m} \left[ I_{mu} + \alpha_c^m \rho_c^m I_{ma} + (\alpha_c^m \rho_c^m + \alpha_t^m \rho_t^m) T_{mu} \right] \right) (S_{fu} + \alpha_{ht}^m S_{fa}) \\[2ex]
0 \\[2ex]
\left( \dfrac{c_m \gamma_m}{N_f} \left[ I_{fu} + \alpha_c^f \rho_c^f I_{fa} + (\alpha_c^f \rho_c^f + \alpha_t^f \rho_t^f) T_{fu} \right] \right) S_{mu} \\[2ex]
\rho_{ht}^f \left( \dfrac{c_m \gamma_m}{N_f} \left[ I_{fu} + \alpha_c^f \rho_c^f I_{fa} + (\alpha_c^f \rho_c^f + \alpha_t^f \rho_t^f) T_{fu} \right] \right) (S_{mu} + \alpha_{ht}^f S_{ma}) \\[2ex]
0
\end{bmatrix}
$$

and

$$
V = - \begin{bmatrix}
-\mu_{f3} I_{fu} \\[1.5ex]
\rho_{ht}^f I_{fu} - \mu_{f4} I_{fa} \\[1.5ex]
\rho_{ct}^f I_{fa} + \rho_{ct}^f T_{fa} - \mu_{f5} T_{fu} \\[1.5ex]
-\mu_{m3} I_{mu} \\[1.5ex]
\rho_{ht}^m I_{mu} - \mu_{m4} I_{ma} \\[1.5ex]
\rho_{ct}^m I_{ma} + \rho_{ct}^m T_{ma} - \mu_{m5} T_{mu}
\end{bmatrix}.
$$

$\mathcal{F}$ and $\mathcal{V}$ are computed as:

$$
\mathcal{F} = \left[ \frac{\partial F_i(x_0)}{\partial x_j} \right] \quad and \quad \mathcal{V} = \left[ \frac{\partial V_i(x_0)}{\partial x_j} \right] \tag{5}
$$

where $x_0$ is the disease free state. Evaluating $\mathcal{F}\mathcal{V}^{-1}$ yields the next generation matrix for the model system (2) whose largest non-negative eigenvalue is the reproduction number, $\mathcal{R}_c$.

$\mathcal{F}\mathcal{V}^{-1}$ and $\mathcal{R}_c$ are given as follows:

$$\mathcal{F}\mathcal{V}^{-1} = \begin{bmatrix} 0 & 0 & 0 & \omega_1\,\eta_1 & \omega_1\,\eta_2 & \omega_1\,\eta_3 \\ 0 & 0 & 0 & \omega_2\,\eta_1 & \omega_2\,\eta_2 & \omega_2\,\eta_3 \\ 0 & 0 & 0 & 0 & 0 & 0 \\ \omega_3\,\varepsilon_1 & \omega_3\,\varepsilon_2 & \omega_3\,\varepsilon_3 & 0 & 0 & 0 \\ \omega_4\,\varepsilon_1 & \omega_4\,\varepsilon_2 & \omega_4\,\varepsilon_3 & 0 & 0 & 0 \\ 0 & 0 & 0 & 0 & 0 & 0 \end{bmatrix}, \tag{6}$$

$$\mathcal{R}_c = \sqrt{\mathcal{R}_{uf}\,\mathcal{R}_{um} + \mathcal{R}_{af}\,\mathcal{R}_{am} + \mathcal{R}_{uf}\,\mathcal{R}_{am} + \mathcal{R}_{af}\,\mathcal{R}_{um}} \tag{7}$$

with

$$\begin{cases} \mathcal{R}_{uf} = \omega_1\,\epsilon_1, \quad \mathcal{R}_{um} = \omega_3\,\eta_1, \quad \mathcal{R}_{af} = \omega_2\,\epsilon_2, \quad \mathcal{R}_{am} = \omega_4\,\eta_2, \\[4pt] \mathcal{R}_u = \mathcal{R}_{uf}\,\mathcal{R}_{um}, \quad \mathcal{R}_a = \mathcal{R}_{af}\,\mathcal{R}_{am}, \quad \mathcal{R}_{mm} = \mathcal{R}_{uf}\,\mathcal{R}_{am}, \quad \mathcal{R}_{mf} = \mathcal{R}_{af}\,\mathcal{R}_{um}, \\[4pt] \omega_1 = \dfrac{c_f\,\gamma_f\,S_{fu}^0}{S_{mu}^0 + S_{ma}^0}, \quad \omega_2 = \dfrac{\rho_{ht}^m\,c_f\,\gamma_f\,(S_{fu}^0 + \alpha_{ht}^m\,S_{fa}^0)}{S_{mu}^0 + S_{ma}^0}, \\[10pt] \omega_3 = \dfrac{c_m\,\gamma_m\,S_{mu}^0}{S_{fu}^0 + S_{fa}^0}, \quad \omega_4 = \dfrac{\rho_{ht}^f\,c_m\,\gamma_m\,(S_{mu}^0 + \alpha_{ht}^f\,S_{ma}^0)}{S_{fu}^0 + S_{fa}^0}, \\[10pt] \eta_1 = \dfrac{1}{\mu_{m3}} + \dfrac{\alpha_c^m\rho_c\rho_{ht}^m}{\mu_{m3}\,\mu_{m4}} + \dfrac{(\alpha_c^m\rho_c + \alpha_t^m\rho_t)\rho_{ct}^m\rho_{ht}^m}{\mu_{m3}\,\mu_{m4}\,\mu_{m5}}, \\[10pt] \eta_2 = \dfrac{\alpha_c^m\rho_c}{\mu_{m4}} + \dfrac{(\alpha_c^m\rho_c + \alpha_t^m\rho_t)\rho_{ct}^m}{\mu_{m4}\,\mu_{m5}}, \quad \eta_3 = \dfrac{(\alpha_c^m\rho_c + \alpha_t^m\rho_t)}{\mu_{m5}}, \\[10pt] \varepsilon_1 = \dfrac{1}{\mu_{f3}} + \dfrac{\alpha_c^f\rho_c\rho_{ht}^f}{\mu_{f3}\,\mu_{f4}} + \dfrac{(\alpha_c^f\rho_c + \alpha_t^f\rho_t)\rho_{ct}^f\rho_{ht}^f}{\mu_{f4}\,\mu_{f3}\,\mu_{f5}}, \\[10pt] \varepsilon_2 = \dfrac{\alpha_c^f\rho_c}{\mu_{f4}} + \dfrac{(\alpha_c^f\rho_c + \alpha_t^f\rho_t)\rho_{ct}^f}{\mu_{f4\,\mu_{f5}}}, \quad \varepsilon_3 = \dfrac{(\alpha_c^f\rho_c + \alpha_t^f\rho_t)}{\mu_{f5}}. \end{cases} \tag{8}$$

$\mathcal{R}_{uf}$, $\mathcal{R}_{um}$ gives the average number of the newly infected unaware AGYW and ABYM whereas $\mathcal{R}_{af}$, $\mathcal{R}_{am}$ gives the average number of the newly infected aware AGYW and ABYM. Newly infected youth generated by individuals with same status is given by $\mathcal{R}_{uf}\,\mathcal{R}_{um}$ and $\mathcal{R}_{af}\,\mathcal{R}_{am}$ whereas newly infected youth generated by mixed status interaction is given by $\mathcal{R}_{uf}\,\mathcal{R}_{am}$ and $\mathcal{R}_{af}\,\mathcal{R}_{um}$. In the absence of HIV testing, condom use and ART control, the control reproduction number $\mathcal{R}_c$ reduces to the basic reproduction number $\mathcal{R}_0$ and this is given as:

$$\mathcal{R}_0 = \sqrt{\mathcal{R}_{0f}\,\mathcal{R}_{0m}} \tag{9}$$

with

$$\mathcal{R}_{0f} = \frac{c_f\, \gamma_f\, S_{fu}^0}{\mu_{f3}\, (S_{mu}^0 + S_{ma}^0)} \quad \text{and} \quad \mathcal{R}_{0m} = \frac{c_m\, \gamma_m\, S_{mu}^0}{\mu_{m3}(S_{fu}^0 + S_{fa}^0)}.$$

## 3 Results

### 3.1 Control reproduction number simulations

Using the parameter estimates for our model system given in Tables 6, 7 and 8, $\mathcal{R}_0$ is estimated at 20.4409 with $\mathcal{R}_{0f} = 22.9550$ and $\mathcal{R}_{0m} = 18.2021$. $\mathcal{R}_{0f} > \mathcal{R}_{0m}$ implies that the adolescent girls and young women have a greater susceptibility to HIV/AIDS infection

**Table 6. Parameter values.**

| Parameter | Value | Unit | Source |
|---|---|---|---|
| $\Lambda_{mu}$, $\Lambda_{ma}$ | 60.7685, 100.9858 | $year^{-1}$ | Data Estimated |
| $\mu_m$ | 0.0101 | $year^{-1}$ | Data Estimated |
| $\tilde{\gamma}_m$ | 2.617 | $year^{-1}$ | Data Estimated |
| $\delta_m$ | 0.0090 | $year^{-1}$ | Data Estimated |
| $\Lambda_{fu}$, $\Lambda_{fa}$ | 61.1842, 118.1215 | $year^{-1}$ | Data Estimated |
| $\mu_f$ | 0.0004 | $year^{-1}$ | Data Estimated |
| $\tilde{\gamma}_f$ | 3.97580754 | $year^{-1}$ | Data Estimated |
| $\delta_f$ | 0.0285 | $year^{-1}$ | Data Estimated |
| $\sigma$ | 0.041667 | $year^{-1}$ | Calculated |
| $\rho_{ht}$ | 0.48 | $year^{-1}$ | Data Estimated |
| $\rho_c$ | 0.3 | $year^{-1}$ | Data Estimated |
| $\rho_t$ | 0.1 | $year^{-1}$ | Data Estimated |

**Table 7. Estimated negative/positive attitude rates towards HIV/AIDS control measures for low control simulations.**

| Parameter | Value | Unit | Source |
|---|---|---|---|
| $\alpha_{ht}^m$, $\alpha_c^m$, $\alpha_t^m$ | 0.15, 0.36, 0.38 | $year^{-1}$ | Data Estimated |
| $\alpha_{ht}^{m1}$, $\alpha_c^{m1}$, $\alpha_t^{m1}$ | 0.99, 0.95, 0.95 | $year^{-1}$ | Data Estimated |
| $\alpha_{ht}^f$, $\alpha_c^f$, $\alpha_t^f$ | 0.25, 0.2, 0.1 | $year^{-1}$ | Data Estimated |
| $\alpha_{ht}^{f1}$, $\alpha_c^{f1}$, $\alpha_t^{f1}$ | 0.97, 0.8, 0.8 | $year^{-1}$ | Data Estimated |

**Table 8. Estimated negative/positive attitude rates towards HIV/AIDS control measures for high control simulations.**

| Parameter | Value | Unit | Source |
|---|---|---|---|
| $\alpha_{ht}^m$, $\alpha_c^m$, $\alpha_t^m$ | 0.1, 0.1, 0.1 | $year^{-1}$ | Assumed |
| $\alpha_{ht}^{m1}$, $\alpha_c^{m1}$, $\alpha_t^{m1}$ | 0.9, 0.9, 0.9 | $year^{-1}$ | Assumed |
| $\alpha_{ht}^f$, $\alpha_c^f$, $\alpha_t^f$ | 0.1, 0.1, 0.1 | $year^{-1}$ | Assumed |
| $\alpha_{ht}^{f1}$, $\alpha_c^{f1}$, $\alpha_t^{f1}$ | 0.9, 0.9, 0.9 | $year^{-1}$ | Assumed |

compared to their male counterparts which is consistent with Kenyan youth HIV/AIDS disease dynamics [1]. The Kenyan reproduction number $\mathcal{R}_0$ was derived from early prevalence antenatal clinic data which was estimated at 6.34 [56]. The presence of combination control measures, however low, has played a key role in reducing new HIV infections among the youthful population with our model control reproduction number $\mathcal{R}_c$ estimated at 4.1003 when $\rho_{ht} = 0.48$, $\rho_c = 0.3$ and $\rho_t = 0.1$ and control attitude rates for the low control simulations given in Table 7.

Fig 6(a)–6(f)) show the change in control reproduction number with fixed HIV/AIDS control measures and varying HIV/AIDS control measures. The control measures are varied from an estimated baseline rate to a 90% efficacy rate. Fig 6(a) 6(b)) show the change in the local control reproduction number when HIV testing is fixed at 0.48 and 0.9 respectively while condom use and ART adherence rates are varied from 0.3–0.9 and 0.1–0.9 efficacy rates. Similarly, Fig 6(c) and 6(d)) show the change in the local control reproduction number when condom use rate is fixed at 0.3 and 0.9 respectively while HIV testing and ART adherence rates are varied from 0.48–0.9 and 0.1–0.9 efficacy rates. Fig 6(e) and 6(f)) show the change in the local control reproduction number when ART adherence is fixed at 0.1 and 0.9 respectively while HIV testing and condom use rates are varied from 0.48–0.9 and 0.3–0.9 efficacy rates.

Fig 6(b), 6(d) and 6(f) generally reflect the impact of reduced transmission potential of the control reproduction number when fixed control measures are at a high efficacy rate of 0.9. The greatest reduction in the control reproduction number is realized when HIV testing rate is fixed at 0.9 with condom use and ART adherence rates increasing from their respective baseline values to 0.9 efficacy rate (see Fig 6(b)). This suggests that fixed higher HIV testing rates in all populations coupled with increased condom use and ART adherence rates work well to reduce the control reproduction number but not below unity for the Kenyan youth. This implies that the current sexual interactions among the various states will sustain the HIV epidemic even when efficacy rate of 90% is achieved.

Taking the best scenario of reduced transmission potential of the control reproduction number described earlier, we unpack the unitary contributors to the control reproduction number to find the best case scenarios that could significantly reduce the control reproduction number (see Fig 7). $\mathcal{R}_u$ contribution will sustain HIV/AIDS at endemic levels among the Kenyan youth population whereas $\mathcal{R}_a$ contribution will result in significant disease reduction among the AGYW and ABYM populations (see Fig 7(a) and 7(b)). Further, any interaction between aware male/female youth with unaware male/female youth yields good result that could lead to significant disease reduction among the Kenyan youth (see Fig 7(c) and 7(d)). Mixed status sexual interaction brings the control reproduction number down in our model as a result of HIV/AIDS status disclosure by the aware AGYW/ABYM. Any sexual relationship fostered with HIV/AIDS tested youth using condoms and adherent to ART promises hope for new HIV/AIDS infection reduction among the Kenyan youth.

### 3.2 Data fitting and parameter estimation

The UNAIDS Kenyan data for HIV/AIDS prevalence was used to fit the AGYW and ABYM model prevalence for both the sex-structured formulation described in section 2.2 and the single-sex formulation given in section 3.2.1. We considered the gender-wise annual HIV prevalence data for the years 1990 to 2018. Table 1 gives the UNAIDS HIV prevalence data summary for the AGYW and ABYM populations respectively [47].

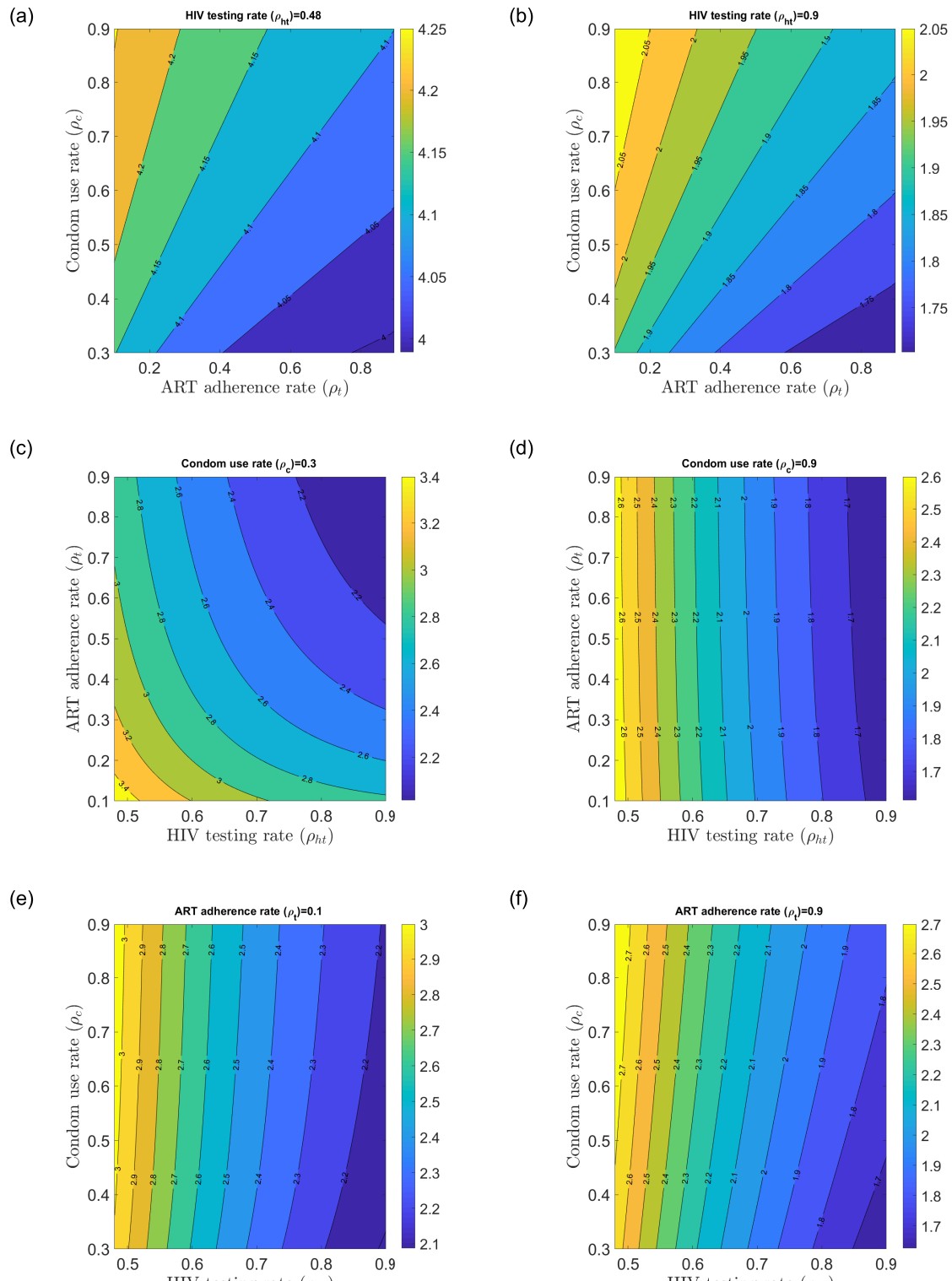

**Fig 6. Change in the local control reproduction number $\mathcal{R}_c$ with varying $\rho_{ht}$, $\rho_c$ and $\rho_t$.** (a) Change in $\mathcal{R}_c$ with low $\rho_{ht}$ and varying $\rho_c$ and $\rho_t$, (b) Change in $\mathcal{R}_c$ with high $\rho_{ht}$ and varying $\rho_c$ and $\rho_t$, (c) Change in $\mathcal{R}_c$ with low $\rho_c$ and varying $\rho_{ht}$ and $\rho_t$, (d) Change in $\mathcal{R}_c$ with high $\rho_c$ and varying $\rho_{ht}$ and $\rho_t$, (e) Change in $\mathcal{R}_c$ with low $\rho_t$ and varying $\rho_{ht}$ and $\rho_c$, (f) Change in $\mathcal{R}_c$ with high $\rho_t$ and varying $\rho_{ht}$ and $\rho_c$.

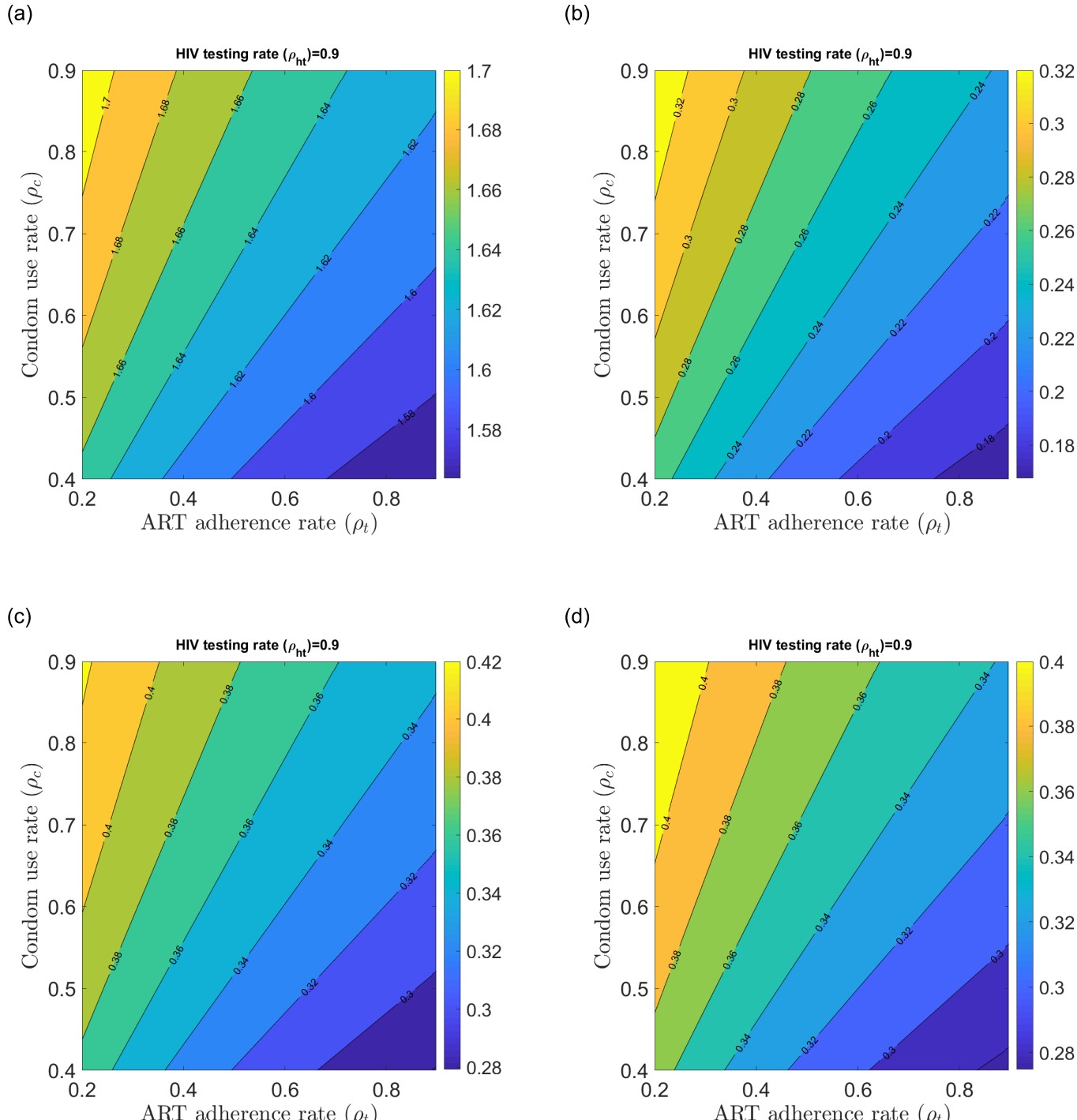

**Fig 7. Change in $\mathcal{R}_u$, $\mathcal{R}_a$, $\mathcal{R}_{mf}$ and $\mathcal{R}_{mm}$ with fixed $\rho_{ht}$ = 0.9 and varying $\rho_c$ and $\rho_t$.** (a) Change in $\mathcal{R}_u$ with high $\rho_{ht}$ and varying $\rho_c$ and $\rho_t$, (b) Change in $\mathcal{R}_a$ with high $\rho_{ht}$ and varying $\rho_c$ and $\rho_t$, (c) Change in $\mathcal{R}_{mf}$ with high $\rho_{ht}$ and varying $\rho_c$ and $\rho_t$, (d) Change in $\mathcal{R}_{mm}$ with high $\rho_{ht}$ and varying $\rho_c$ and $\rho_t$.

We define the AGYW and ABYM model prevalence as follows:

$$\text{AGYW Model Prevalence} = \frac{\text{Total number of infected AGYW}}{\text{Total AGYW population}} = \frac{I_{fu} + I_{fa} + T_{fu}}{N_f}, \quad (10)$$

$$\text{ABYM Model Prevalence} = \frac{\text{Total number of infected ABYM}}{\text{Total ABYM population}} = \frac{I_{mu} + I_{ma} + T_{mu}}{N_m}. \quad (11)$$

The AGYW and ABYM model prevalence described in Eqs 10 and 11 are fitted to the UNAIDS HIV prevalence data given in Table 1 to estimate parameters in Tables 6 and 7. Using MATLAB built in functions 'ODE45' and 'fminsearch' we estimated the parameters in Tables 6 and 7 by minimizing the sum of square difference of the AGYW and ABYM model prevalence solution and the HIV prevalence data for the AGYW and ABYM populations given in Eqs 12 and 13 as

$$SS^f = \sum_{k=1}^{29} \left( \frac{\left[ \frac{I_{fu}^k + I_{fa}^k + T_{fu}^k}{S_{fu}^k + S_{fa}^k + I_{fu}^k + I_{fa}^k + T_{fu}^k + T_{fa}^k} - \tilde{Q}_1^k \right]^2}{[Max(\tilde{Q}_2^k, \tilde{Q}_3^k)]^2} \right), \quad (12)$$

$$SS^m = \sum_{k=1}^{29} \left( \frac{\left[ \frac{I_{mu}^k + I_{ma}^k + T_{mu}^k}{S_{mu}^k + S_{ma}^k + I_{mu}^k + I_{ma}^k + T_{mu}^k + T_{ma}^k} - \tilde{Q}_4^k \right]^2}{[Max(\tilde{Q}_5^k, \tilde{Q}_6^k)]^2} \right). \quad (13)$$

To estimate parameters with little uncertainty, the 'fminsearch' algorithm in MATLAB software computes the goodness of fit by calculating the minimum sum of squares due to error (SSE). The minimum value of sum of squares due to error that is closer to 0 implies that the model has a smaller random error component and the resulting fit can be used for prediction [57]. This approach of fitting has also been used successfully elsewhere [58–60]. The higher the minimum value of SSE, the greater the variation from the prevalence data. For our model fit, the SSE prevalence fit for the AGYW was found to be 0.0167 whereas that for the ABYM was 0.0450. Given that the minimum SSE values we obtained are close to 0, the estimated parameters can be trusted and used for the time series model simulations.

The time length for the years 1990 to 2018 is given as $k$ with $\tilde{Q}_1^k$, $\tilde{Q}_4^k$ being the yearly AGYW/ABYM UNAIDS prevalence data, $\tilde{Q}_2^k$, $\tilde{Q}_5^k$ the maximum yearly AGYW/ABYM model prevalence solutions and $\tilde{Q}_3^k$, $\tilde{Q}_6^k$ the maximum yearly AGYW/ABYM UNAIDS prevalence data. $S_{fu}^k$, $S_{fa}^k$, $I_{fu}^k$, $I_{fa}^k$, $T_{fu}^k$, $T_{fa}^k$, $S_{mu}^k$, $S_{ma}^k$, $I_{mu}^k$, $I_{ma}^k$, $T_{mu}^k$, $T_{ma}^k$ are numerically computed solutions at each time $k$.

Attitudes affecting efficacy of HIV testing rate $\rho_{ht}$, condom use rate $\rho_c$ and ART adherence rate $\rho_t$ negatively $\alpha_{ht}^f$, $\alpha_c^f$, $\alpha_t^f$, $\alpha_{ht}^m$, $\alpha_c^m$, $\alpha_t^m$ and positively $\alpha_{ht}^{f1}$, $\alpha_c^{f1}$, $\alpha_t^{f1}$, $\alpha_{ht}^{m1}$, $\alpha_c^{m1}$, $\alpha_t^{m1}$ are estimated whereas the exit parameter $\sigma$ is calculated as 1/24 given that the AGYW and ABYM exit the model at the age of 24 years. The best parameters estimated by model fitting and calculated parameter are given in Tables 6 and 7 with $\tilde{\gamma}_f = c_f \gamma_f$ and $\tilde{\gamma}_m = c_m \gamma_m$.

We used the 2012 KAIS data described in section 2.1.1 to estimate the initial population for the state variables $S_{fu}(0) = 636$, $S_{fa}(0) = 1006$, $T_{fa}(0) = 5$, $S_{mu}(0) = 694$, $S_{ma}(0) = 867$ and $T_{ma}(0) = 3$. We estimated the initial infected population for our model as $I_{fu}(0) = 54$, $I_{fa}(0) = 76$, $T_{fu}(0) = 10$, $I_{mu}(0) = 13$, $I_{ma}(0) = 26$ and $T_{mu}(0) = 5$.

In the absence of control measures, the Kenyan youth model prevalence trends increases with time (see Fig 8(a) and 8(b)). Interestingly, the ABYM model prevalence exceeds the AGYW model prevalence when intervention is absent (see Fig 8(a) and 8(b)). The Kenyan youth model prevalence without control measures only fits the initial rise of the HIV/AIDS epidemic.

While the earliest cases of HIV/AIDS in Kenya were reported in the 1980's, it was only until the late 1990's that the HIV/AIDS epidemic increased from 5.3% in 1990 to a peak prevalence of 10.5% in the years 1995-1996 and by 2003, the HIV/AIDS prevalence had declined to about 6.7% [61]. A combination of factors such as higher mortality rates, sexual behaviour change, lower incidences, delay in sexual debut among others contributed to the dramatic decline in Kenya's HIV/AIDS epidemic [61]. It is possible that even the Kenyan youth adopted safer sexual behaviors including condom use, reduction of multiple sexual partners and delay in first sex. Thus, fitting the AGYW and ABYM model prevalence to the Kenyan youth UNAIDS HIV/AIDS data subject to the estimated HIV testing, condom use and ART adherence control measures with disproportional AGYW/ABYM attitudes affecting the mentioned control measures efficacy resulted in a good fit (see Fig 9(a) and 9(b)).

AGYW HIV/AIDS model prevalence fits well to the Kenyan UNAIDS female youth HIV/AIDS prevalence when negative attitudes towards HIV testing, condom use and ART adherence are lower in AGYW population at 18% and higher in ABYM population at 30% with positive attitudes towards the three HIV/AIDS control measures greater in AGYW population at 86% compared to ABYM population which is at 69%. Similarly, ABYM model prevalence fits well when negative attitudes towards HIV/AIDS control measures are greater in AGYW population at 33.7% and positive attitudes greater in ABYM population at 96%. Our results project a decrease in the AGYW prevalence trend from 2.5 in 2018 to about 2.17745 in 2023 (see Figs 4(a) and 9(a)). Similarly, our model predicts a decrease in the ABYM prevalence trend from 1.5 in 2018 to about 1.44855 in 2023 (see Figs 4(b) and 9(b)). These results hold assuming the control measures and the constant negative/positive attitudes towards the control measures remain the same.

We used the parameter values given in Table 6 to perform the numerical simulations for the model system (2) and the control reproduction number in section 2.4 with low control attitude rates given in Table 7 and high control attitude rates given in Table 8.

**3.2.1 Single-sex youth model fit.** We considered the single-sex youth model given in model system (14) to understand factors influencing its model fit. The incidence rates $\beta_u$, $\beta_a$, $\tilde{\beta}_a$ and exit rates $\mu_1, \mu_2, \ldots, \mu_6$ are given in equation 16 in S2 Appendix. See S1 and S2 Tables for the single-sex model state variables and parameters description.

$$
\begin{cases}
\dfrac{dS_u}{dt} = \Lambda_u - \beta_u S_u - \beta_a S_u - \mu_1 S_u, \\[2mm]
\dfrac{dS_a}{dt} = \Lambda_a + \rho_{ht} S_u - \tilde{\beta}_a S_a - \mu_2 S_a, \\[2mm]
\dfrac{dI_u}{dt} = \beta_u S_u - \mu_3 I_u, \\[2mm]
\dfrac{dI_a}{dt} = \tilde{\beta}_a S_a + \beta_a S_u + \rho_{ht} I_u - \mu_4 I_a, \\[2mm]
\dfrac{dT_u}{dt} = \rho_{ct} I_a + \rho_{ct} T_a - \mu_5 T_u, \\[2mm]
\dfrac{dT_a}{dt} = \rho_{ct}^1 I_a + \rho_{ct}^1 T_u - \mu_6 T_a.
\end{cases}
\tag{14}
$$

(a)

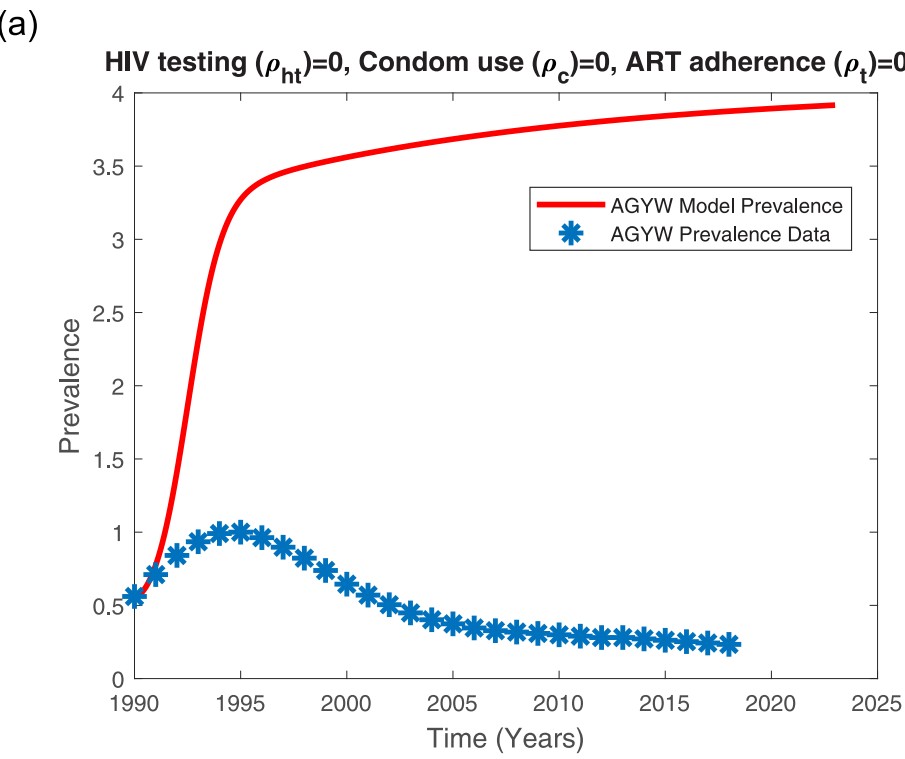

(b)

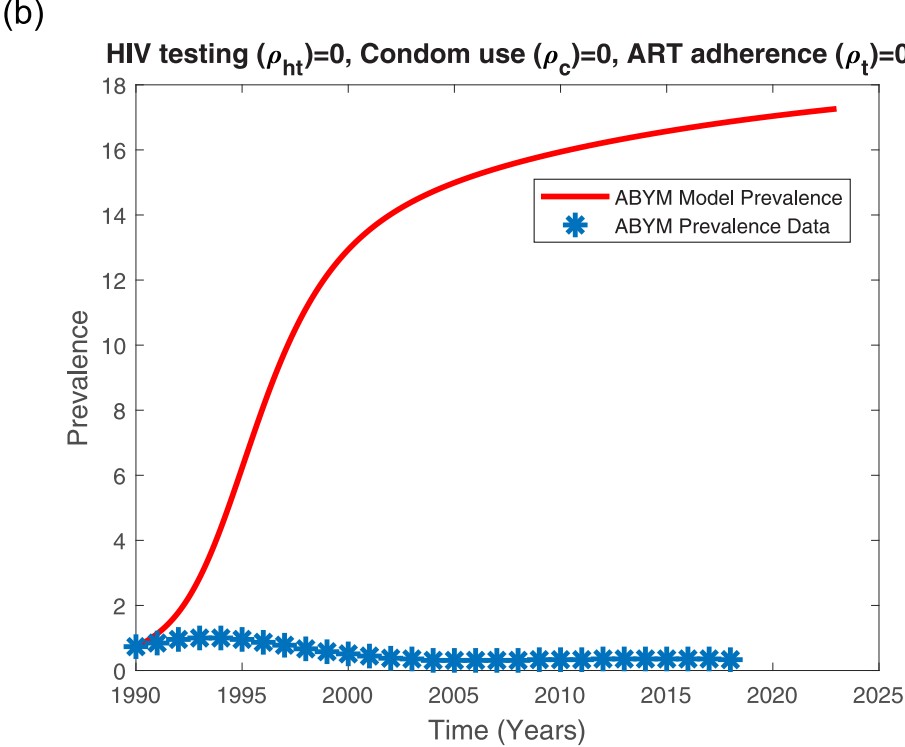

**Fig 8. AGYW and ABYM model prevalence with no control fitted to UNAIDS AGYW and ABYM prevalence data respectively.** (a) AGYW model prevalence with no control, (b) ABYM model prevalence with no control.

(a)

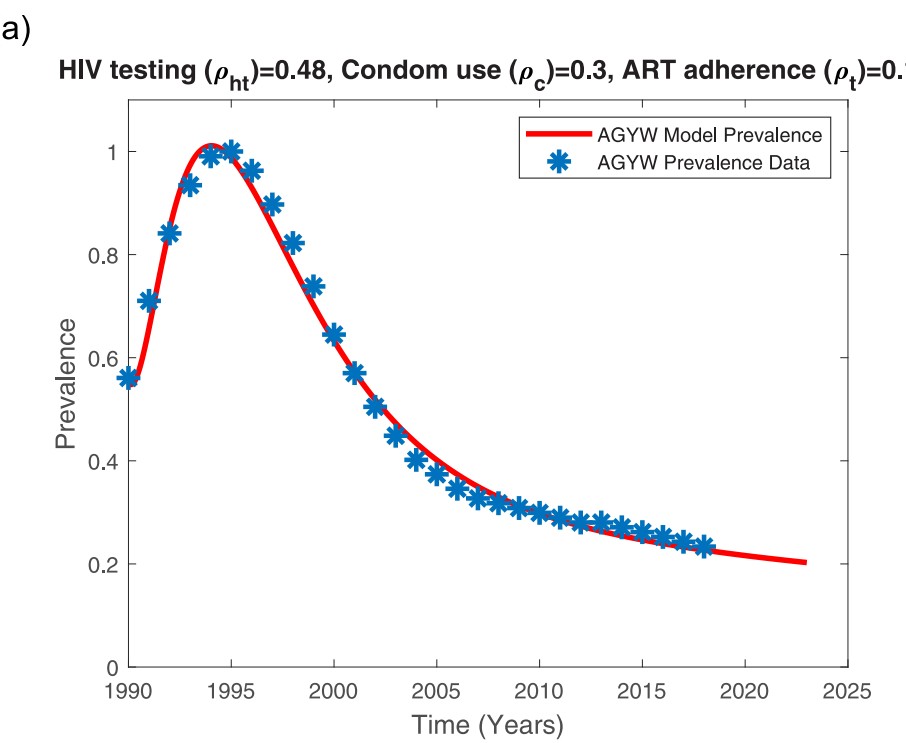

(b)

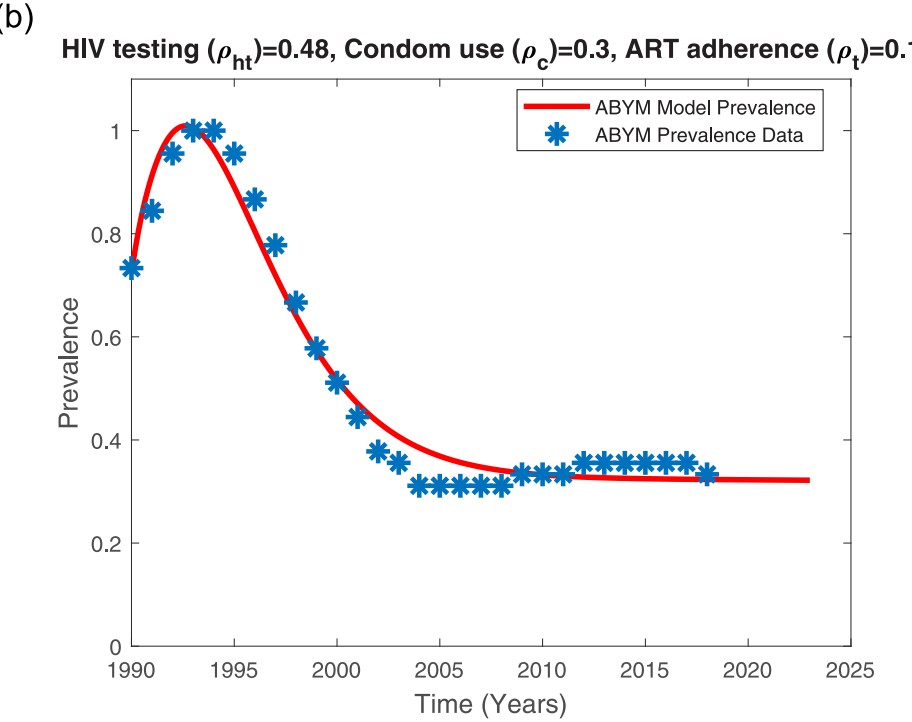

**Fig 9. AGYW and ABYM model prevalence with low control fitted to UNAIDS AGYW and ABYM prevalence data respectively.** (a) AGYW model prevalence with low control, (b) ABYM model prevalence with low control.

We fitted the single-sex model to the averaged AGYW/ABYM UNAIDS-Kenya HIV/AIDS prevalence data given in Table 1. Using AGYW/ABYM averaged initial conditions in section 3.2 and parameter values given in S3 Table yields the model fit given in Fig 10(a). Adjusting the transmission risk and contact rates (see S4 Table) results in a good fit (see Fig 10(b)).

The single sex-structured model fits well to data (SSE = 0.0232) when HIV testing rate, condom use rate and ART adherence rates are 0.48, 0.3 and 0.1 respectively with the product of probability of transmission risk ($\gamma$) and contact rate ($c$) reduced from 3.17245525 to 0.03022869. This seems to suggest that for the single-sex structured model, change in contact behavior could have influenced the change in HIV/AIDS prevalence trends among the youth. When we reduced the contact rate and probability of transmission risk in the sex-structured model, the resultant prevalence fit was poor and only a good fit was realized when the gender-wise attitudes towards HIV/AIDS control measures were disproportional. The sex-structured model further revealed that disproportional gender-wise attitudes towards HIV/AIDS control measures could have also influenced the Kenyan youth HIV/AIDS prevalence trends.

## 3.3 Model simulations results

Numerical simulations on the model system Eq (2) are carried out to test the AGYW and ABYM HIV/AIDS epidemic behavior. The 2020 UNAIDS 90-90-90 HIV/AIDS eradication plan aims to have at least 90% HIV testing coverage for all persons living with HIV with at least 90% initiated on ART achieving a 90% viral load suppression [19]. This informed the 90% HIV testing and ART efficacy rates for our high control simulations. Male condoms when used correctly and consistently in every sexual intercourse is estimated to have at least 90% efficacy against HIV/AIDS transmission whereas female condoms offer at least 94% protection [62]. In Kenya, male condoms are most preferred as described in section 1. Hence, we used the male condom efficacy of 90% to model high control cases. The baseline rates for HIV testing $\rho_{ht}$ = 0.48, condom use $\rho_c$ = 0.3 and ART adherence $\rho_t$ = 0.1 were estimated by model fitting as described in section 3.2. Estimated constant negative/positive attitudes towards HIV/AIDS control measures for the low control and high control simulations are given in Tables 7 and 8 respectively.

Figs 11(a), 12(a), 13(a) and 14(a) suggest that with time the Kenyan youth HIV/AIDS epidemic matures and attains stability without any intervention. However, the prevalence doesn't decline after attaining stability in the absence of HIV/AIDS control measures (see Fig 14(a)). Low control use ($\rho_{ht}$ = 0.48, $\rho_c$ = 0.3, $\rho_t$ = 0.1) with estimated attitude rates given in Table 7 seems to reduce the infected populations and the AGYW/ABYM model prevalence with better benefits in the ABYM population (see Figs 12(b), 13(b) and 14(b)).

High control rates, $\rho_{ht}$ = 0.9, $\rho_c$ = 0.9, $\rho_t$ = 0.9, with reduced negative control attitudes and increased positive control attitudes in all populations seems to have a significant effect in HIV/AIDS disease decline among the AGYW and ABYM populations as the infected populations are reduced significantly with similar trends observed in the youth prevalence (see Figs 12(c), 13(c) and 14(c)). Interestingly, when the negative attitudes towards condom use and ART adherence among the AGYW and ABYM population are slightly increased when HIV/AIDS control measures are low, the youth HIV/AIDS model prevalence begins to increase despite the initial decline (see Fig 14(d)).

We investigated the effects of varying HIV testing rates, condom use rates and ART adherence rates among the adolescent girls and young women (AGYW) and, adolescent boys and young men (ABYM) populations aged 15-24. We considered constant negative and positive attitudes influencing the uptake of HIV/AIDS control measures in these populations. HIV testing rates, condom use rates and ART adherence rates were varied from their estimated low

(a)

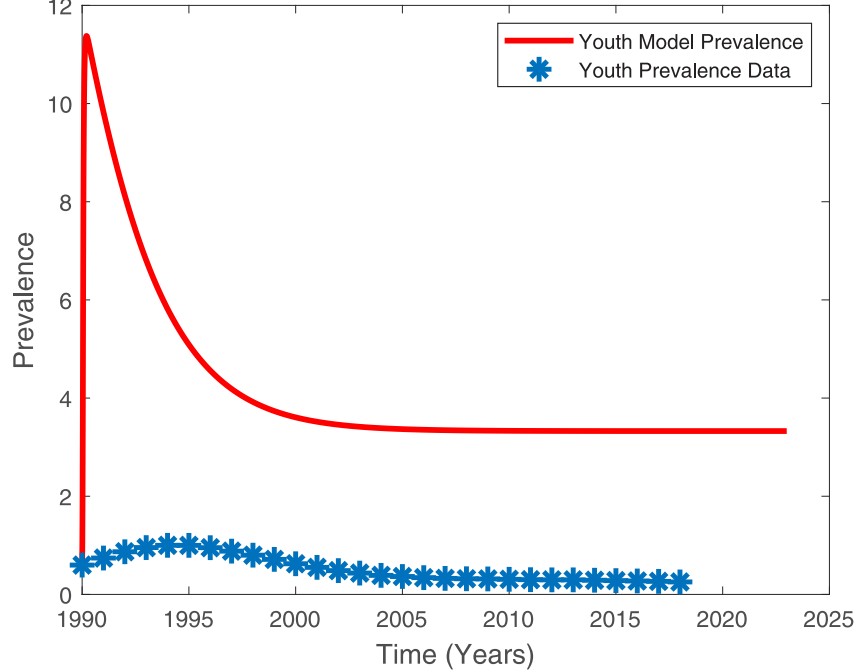

(b)

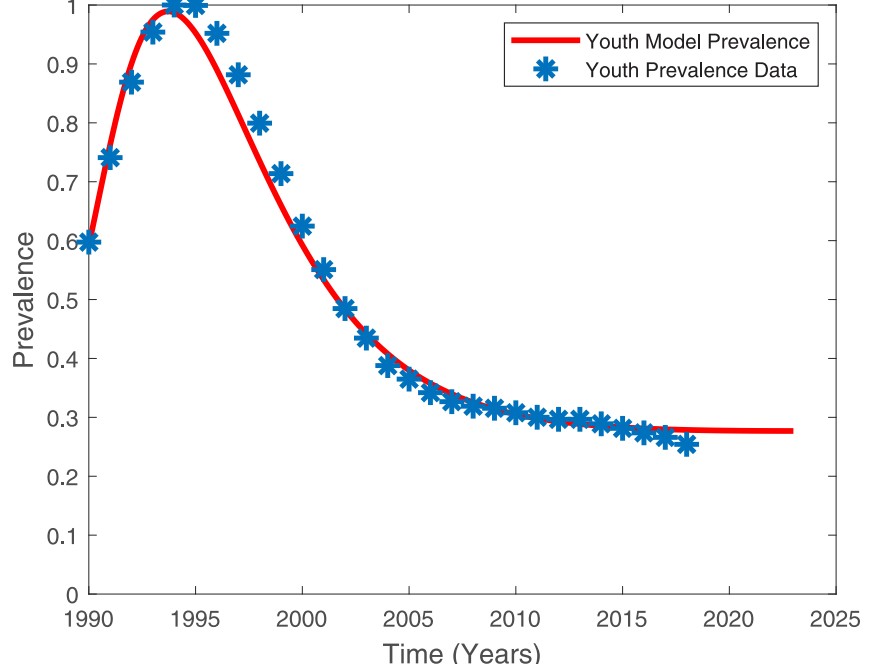

**Fig 10. Single-sex model prevalence with varying transmission risk and contact rate fitted to averaged UNAIDS AGYW and ABYM prevalence data.** (a) Single-sex model prevalence with high transmission risk and high contact rate, (b) Single-sex model prevalence with reduced transmission risk and reduced contact rate.

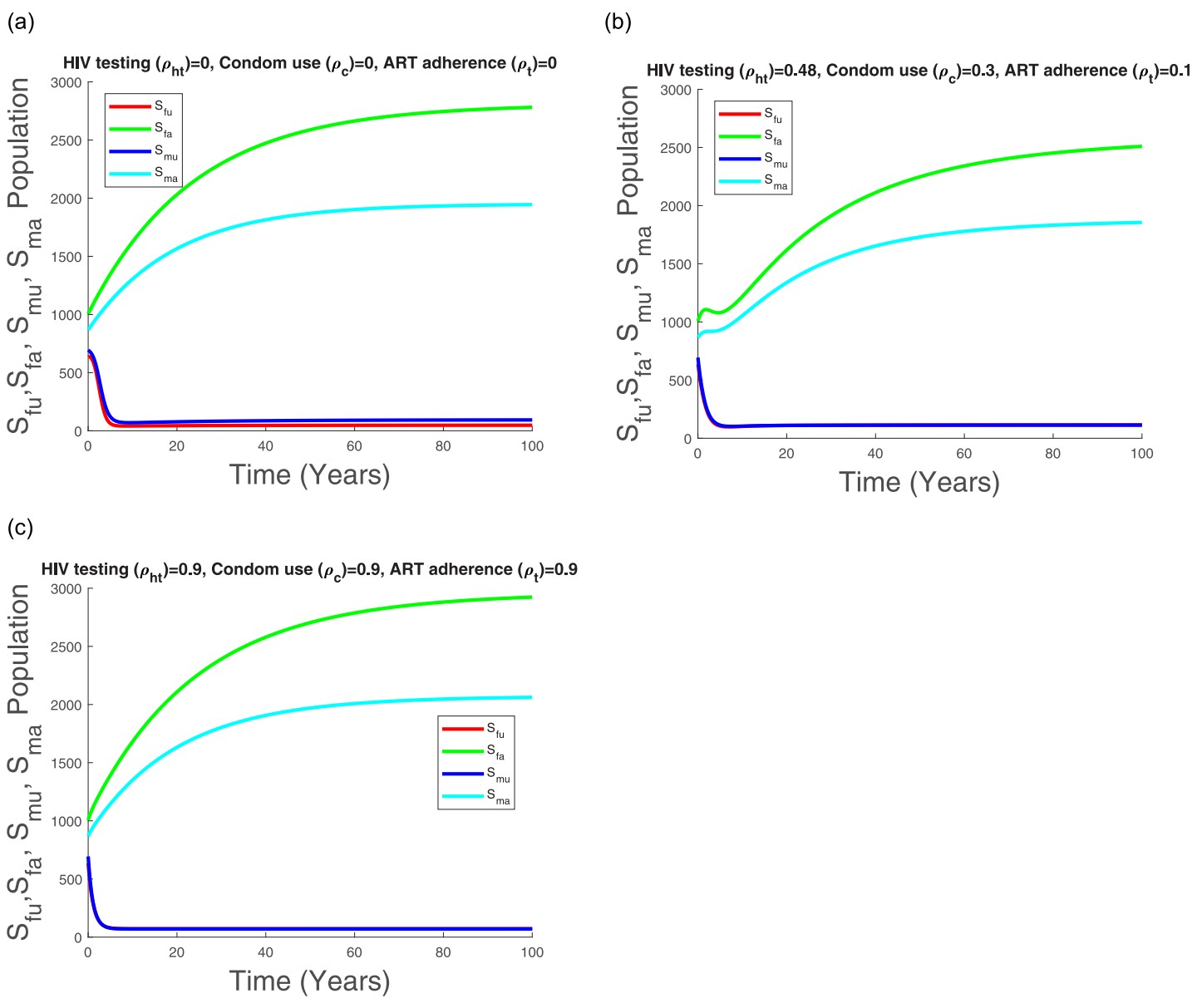

**Fig 11. Transmission dynamics of $S_{fu}$, $S_{fa}$, $S_{mu}$ and $S_{ma}$ populations with varying control.**

baseline rates of 0.48, 0.3, 0.1 respectively to the estimated efficacy rates of 0.9 each. Low control simulations were associated with increased constant negative attitudes towards HIV/AIDS control measures whereas high control simulations were associated with reduced negative attitudes towards HIV/AIDS control measures and increased constant positive attitudes towards HIV/AIDS control measures among the AGYW/ABYM populations and the Kenyan society/cultural groups. The susceptible and infected AGYW/ABYM populations were each differentiated into two broad categories according to their HIV/AIDS status knowledge. That is, uninfected aware or uninfected unaware and infected aware or infected unaware. Infected aware populations were further differentiated into two categories based on their condom use and ART adherence. Unaware populations could change their status and move to aware

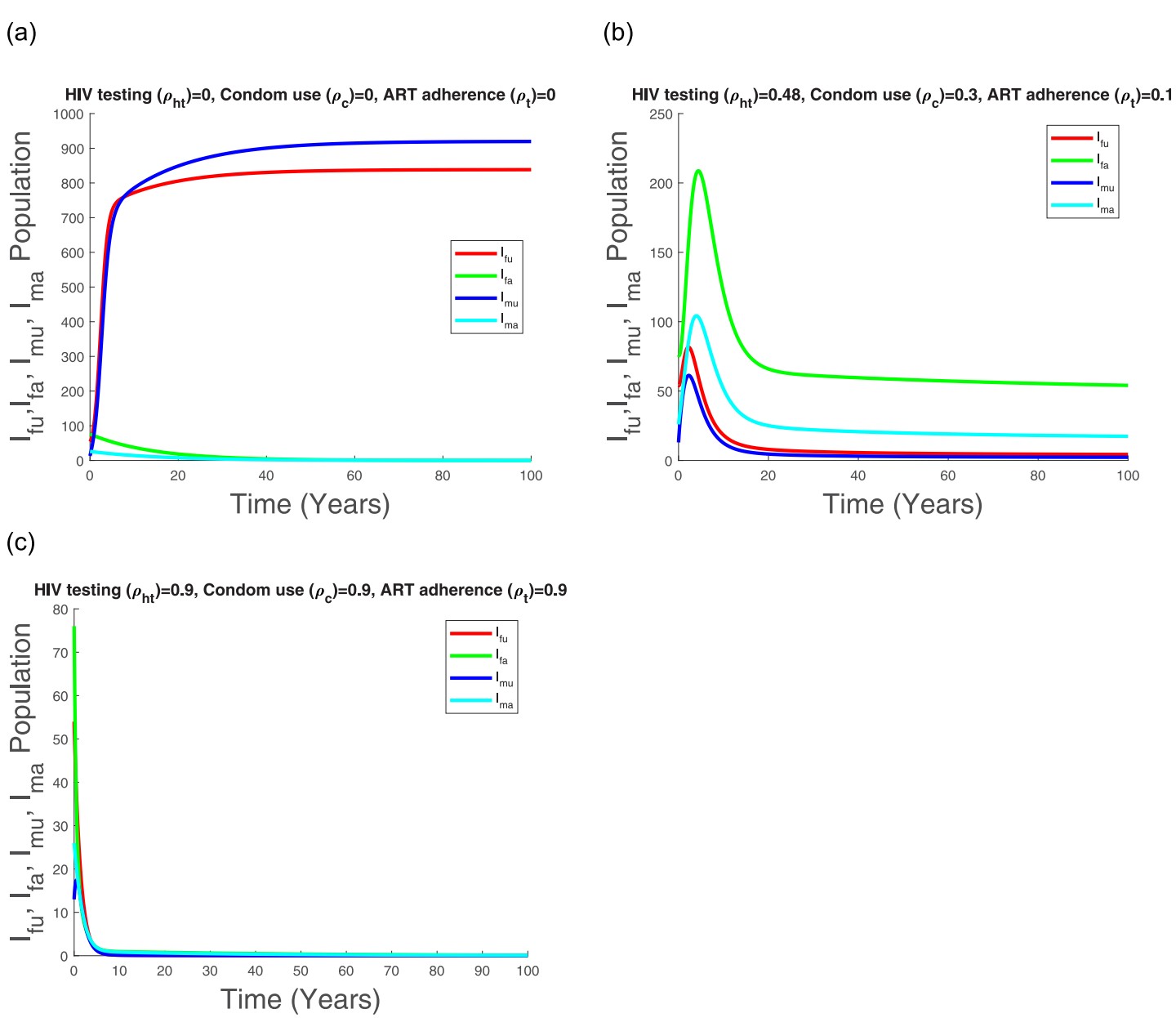

**Fig 12. Transmission dynamics of $I_{fu}$, $I_{fa}$, $I_{mu}$ and $I_{ma}$ population with varying control.**

populations through HIV testing, condom use or ART initiation. This model structure was largely informed by the 2012 Kenya AIDS Indicator Survey (KAIS) [44].

We fitted both the single-sex model and the sex-structured model to UNAIDS-Kenya HIV Surveillance prevalence data for the young males and young females aged 15-24. The sex-structured HIV/AIDS model prevalence seems to fit to each of their estimated UNAIDS-Kenya HIV/AIDS prevalence data reasonably well when negative/positive attitudes towards HIV/AIDS control measures were disproportional in the AGYW/ABYM populations whereas the single-sex model prevalence trend seemed sensitive to transmission risk and contact rate. The single sex-structured model suggests that reduced transmission risk and sexual contact rate in the presence of low control measures could have resulted in reduced HIV/AIDS prevalence

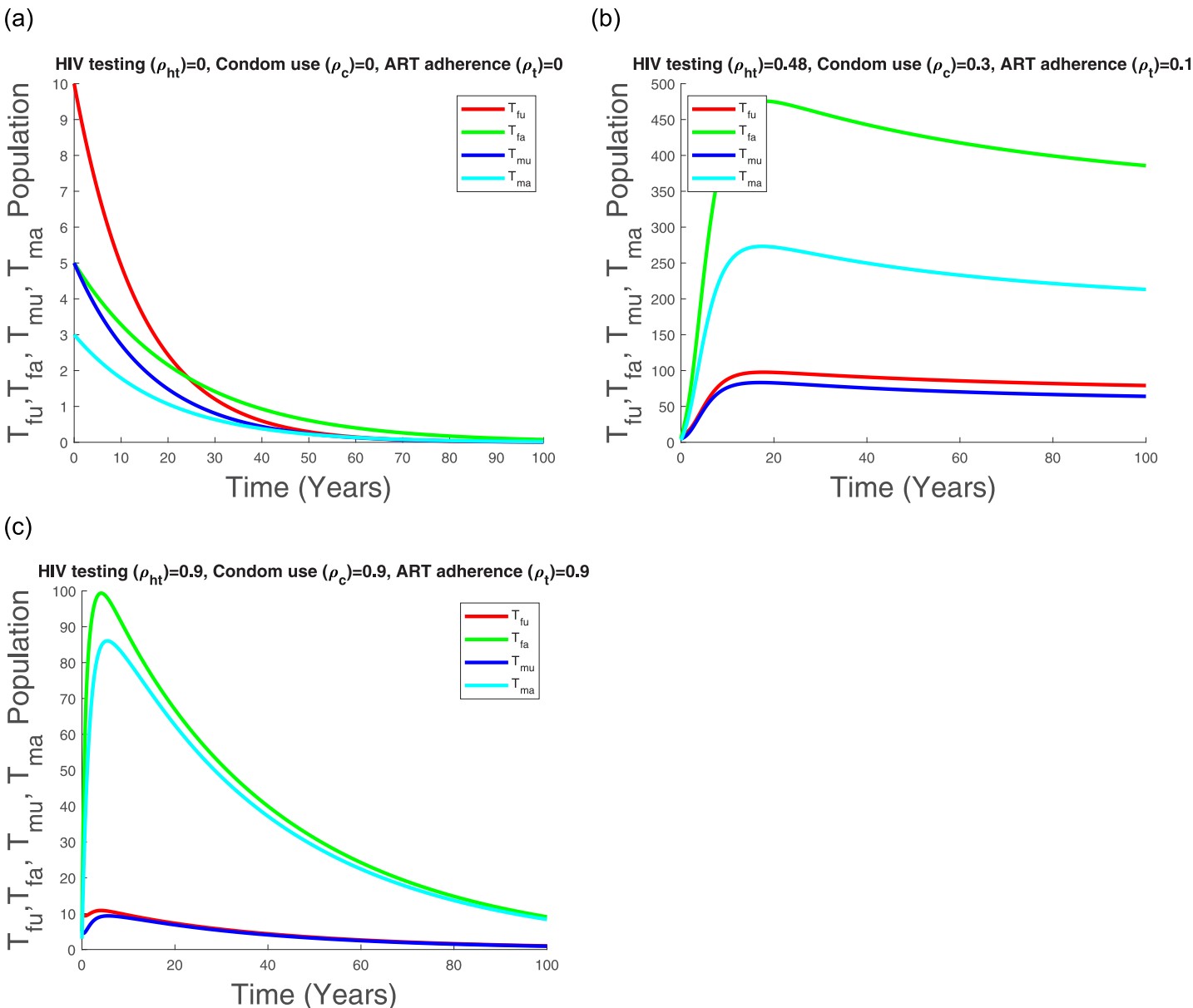

**Fig 13. Transmission dynamics of $T_{fu}$, $T_{fa}$, $T_{mu}$ and $T_{ma}$ population with varying control.**

among the youth in Kenya. The sex-structured model seemed to reveal further the effects of disproportional gender-wise attitudes towards HIV/AIDS control measures affecting uptake of control measures in the youth populations. Increased ABYM infectivity and reduced AGYW infectivity resulted in the female youth model good fit whereas increased AGYW infectivity and reduced ABYM infectivity resulted in the male youth good model fit. In addition to reduced transmission risk and contact rate, it seems that gender-wise attitudes towards HIV/AIDS control measures played a role in reducing HIV/AIDS prevalence among the youth in Kenya. The AGYW/ABYM model fit estimated the best parameters for model simulations.

Simulations on the control reproduction number revealed the impact of reduced transmission potential of the control reproduction number but not below unity when HIV testing rate

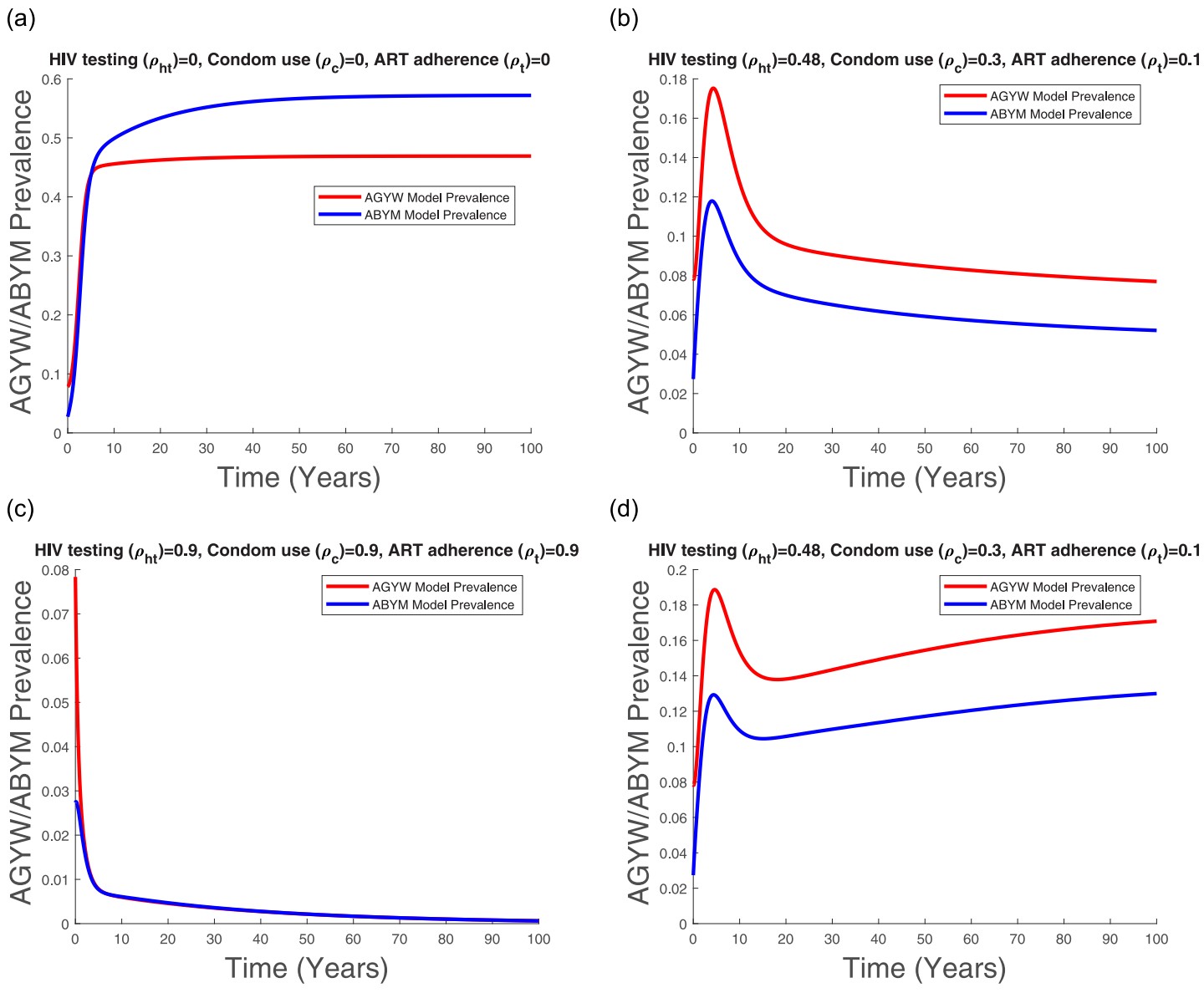

**Fig 14. AGYW and ABYM model prevalence with varying control.** (a) No control, (b) Low control paired with Table 7 attitudes, (c) High control paired with Table 8 attitudes, (d) Low control with increased negative attitudes towards condom use and ART adherence.

was fixed at a high efficacy rate of 0.9 with increasing condom use and ART adherence to high efficacy rates. This was as a result of the complex sexual structure among the Kenyan youth with the HIV/AIDS disease being sustained at endemic levels by the unaware youth. The simulations suggest that significant HIV/AIDS reduction among the Kenyan youth will only be possible if for each sexual relationship established, there is at least one partner who is willing to disclose his/her HIV/AIDS status to his/her sex partner as well as use protection consistently. Numerical simulations on our model system revealed the impact of successful combination control approach in drastically reducing new HIV/AIDS infection. Low combination control approach has a positive effect in reducing youth disease prevalence with better benefits in the ABYM population provided the negative attitudes towards HIV/AIDS control are kept in check. Slight increase in negative attitudes towards AGYW/ABYM condom use or ART

adherence can easily increase the youth disease prevalence even after the initial disease decline. Significant HIV/AIDS disease reduction is achieved only when positive attitudes towards HIV/AIDS control measures are increased in all AGYW/ABYM populations with decreasing negative attitudes.

## 4 Discussion

Globally, male and female youth are central in the HIV/AIDS action plans due to the high numbers of youth unaware of their HIV/AIDS status [2, 63]. The 2012 Kenya AIDS Indicator Survey (KAIS) also revealed a worrying trend of many infected male and female youth unaware of their HIV/AIDS status and this is consistent with the global trends [44, 63]. The social attitudes influencing HIV testing, condom use and ART adherence efficacy cannot be downplayed as they play a critical role in either fueling the HIV/AIDS epidemic or curtailing its spread in this population group as evidenced by the model results. The female youth HIV/AIDS prevalence trend seems to be associated with increased male infectivity with decreased female infectivity while the male youth prevalence trend seems to be associated with increased female infectivity and reduced male infectivity.

The annual increase of new HIV infections among the youth exceeds HIV/AIDS related deaths which in turn increases the net size of HIV/AIDS infected population in the country [45]. This remains a huge concern since, as the HIV/AIDS infected youth population continues to increase, the risk of HIV/AIDS transmission increases too. Kenya's HIV/AIDS response is quite dynamic and there is increased efforts in scaling up HIV testing, condom use and ART adherence among the AGYW and ABYM populations. Our model results reflect the importance of addressing the social attitudes inhibiting efficacy of HIV testing, condom use and ART adherence among the Kenyan youth. While combination control measures play a huge role in reducing HIV/AIDS prevalence trends among the youth in Kenya, the disease may still remain endemic provided the infected unaware populations' sexual interactions exist. Our results suggest that it is necessary to scale up HIV testing among the youth while at the same time addressing factors affecting its efficacy such as perceived individual's risk to HIV infection, HIV/AIDS knowledge, education, inadequate health services among others. It is also necessary to address the societal norms, psycho-social conditions, stigma, socio-cultural factors associated with condom use and ART adherence among the youth in Kenya. Their negative influence is possibly one of the significant drivers for the reversal of decades of successful control measures geared at reducing HIV/AIDS prevalence in Kenya.

The 2014/2015—2018/2019 Kenya AIDS Strategic Framework (KASF) by the Ministry of Health goal was to significantly reduce new HIV infections, AIDS-related mortality, HIV/AIDS related stigma and discrimination and, significantly increase domestic financing of HIV/AIDS response programmes [64]. KASF plan ties together with Kenya's 2030 vision of an economically transformed nation where health plays a key role in realizing this goal. HIV/AIDS epidemic in Kenya significantly increases the disease burden in the country and part of Kenya's Vision 2030 is to have a country free of HIV infections, HIV-related stigma and AIDS-related deaths. Despite the considerable progress in reducing new HIV infections among the youth in Kenya since the KASF initiation, challenges surrounding policy implementation and community response continue to affect effective HIV/AIDS response [65, 66]. The time series model predictions from our study suggested that Kenya's Vision 2030 of a country free of new HIV infections might not be realized given the low HIV/AIDS control measures and societal attitudes hampering the uptake of HIV/AIDS control measures by the youth who are a priority population targeted in Kenya's HIV response.

Since Kenya became a low middle income country in 2014, the progress towards HIV/AIDS control slowed down [64]. The country's income status drastically reduced international donor support on HIV/AIDS policy implementation and monitoring of key prevention areas such as HIV testing, condom use and ART adherence [64, 66]. Reduced HIV/AIDS funding has significantly affected programs addressing social drivers of HIV/AIDS such as societal attitudes, which directly influence the uptake of HIV/AIDS control measures by the youth [65, 66]. Low funding has also affected Community Based Organizations and Community Leadership who play a key role in addressing the societal attitudes directly affecting the uptake of HIV/AIDS control measures hence, increasing the youths vulnerability to HIV infection [64]. Our study suggested that for significant reduction of new HIV infections and possible elimination of new HIV infections among the youth, key intervention areas such as HIV screening, condom use and ART adherence needs to be significantly increased and societal negative attitudes directly affecting the uptake of these control measures significantly reduced. Hence, it will be necessary to address challenges affecting HIV/AIDS funding and empower Community Based Organizations and Community Leadership so as to successfully combat the root cause (societal attitudes) affecting the uptake of HIV/AIDS control measures by the Kenyan youth.

As far as we know, there are no existing mathematical models that have addressed the impact of combination control measures and their influence among the youth HIV/AIDS disease dynamics in Kenya with differentiated HIV/AIDS status knowledge using two-sex structured models. Multiple control strategies such as HIV screening, ARV drug treatment and condom use in a single sex-structured model was considered by [38] to understand the potential impact on the current HIV/AIDS control measures. Their results reflected the projections of HIV/AIDS epidemic trends when HIV/AIDS control measures and multiple sex partners varied. Our work presented similar results using a single-sex structured model which further revealed the effects of transmission risk and contact rate in informing the Kenyan youth HIV/AIDS prevalence trends. The limitations of a single-sex structured model was evident in our work where the single-sex structured model could not fit the HIV/AIDS prevalence when control measures were influenced by gender-wise societal attitudes that were incorporated into the model. The two-sex structured model in this study resolved this weakness. The importance of two-sex model speaks to increased mathematical complexity but provided an appropriate tool to explain the associated drivers of the Kenyan youth HIV/AIDS dynamics.

Having studied the impact of combination control strategies and constant negative/positive attitudes influencing the efficacy of the HIV/AIDS control measures among the youth infected populations in a single patch model, it will be interesting to study the effects of combination control in a metapopulation model in Kenya given that this population group is highly mobile. Dynamic attitudes towards HIV/AIDS control measures should also be considered. We used the UNAIDS-Kenya HIV Surveillance data to fit our model which is not exempt from biases due to insufficient nationally representative HIV/AIDS prevalence data. For accurate model fitting to national prevalence trends, nationally representative HIV/AIDS surveillance need to be increased so as to create a larger prevalence data pool. While this study focused on population dynamics of the AGYW/ABYM, it will be interesting to study the individual based model for this AGYW/ABYM formulation. Given the behavior heterogeneity among the youth, studying each individual behavior explicitly to population level could give deeper insights in understanding the social drivers of HIV among the Kenyan youth. This in turn will help influence relevant policies geared at eradicating new HIV infections among the youth in Kenya.

## Supporting information

**S1 Appendix. Endemic equilibrium expressions.**
(PDF)

**S2 Appendix. Single-sex model description and parameter values.**
(PDF)

**S1 Table. Description of single-sex model state variables.**
(PDF)

**S2 Table. Description of single-sex model parameters.**
(PDF)

**S3 Table. Parameter values for the single-sex model, $\tilde{\gamma} = c\gamma$.**
(PDF)

**S4 Table. Adjusted parameter values for the single-sex model.**
(PDF)

## Acknowledgments

The authors appreciate the support from the Organization for Women in Science for the Developing World (OWSD), the Simons Foundation, Mawazo Institute, University of Nairobi (Kenya) and University of KwaZulu-Natal (South Africa). We thank Mr. Innocent B. Mboya of University of KwaZulu-Natal for his guidance in data analysis.

## Author Contributions

**Conceptualization:** Marilyn Ronoh, Faraimunashe Chirove, Josephine Wairimu, Wandera Ogana.

**Formal analysis:** Marilyn Ronoh.

**Methodology:** Marilyn Ronoh, Faraimunashe Chirove, Josephine Wairimu.

**Supervision:** Faraimunashe Chirove, Josephine Wairimu, Wandera Ogana.

**Writing – original draft:** Marilyn Ronoh.

**Writing – review & editing:** Marilyn Ronoh.

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
