## [Decision Letter · Decision Letter 0]

4 Sep 2020

PONE-D-20-04014

Evidence-based modeling of combinatory control on Kenyan youth HIV/AIDS dynamics

PLOS ONE

Dear Dr. Ronoh,

Thank you for submitting your manuscript to PLOS ONE. After careful consideration, we feel that it has merit but does not fully meet PLOS ONE’s publication criteria as it currently stands. Therefore, we invite you to submit a revised version of the manuscript that addresses the points raised during the review process.

These analyses describe the trajectory of the HIV epidemic among youth in Kenya with various intervention coverages and with the added impact of positive or negative attitudes towards these interventions. Although attitudes can impact adherence to interventions, it is not clear how modeling attitudes and their impact in coverage is different than modeling the impact of varying coverage of these interventions. Furthermore, it is challenging to measure how negative attitudes can affect level of coverage, so the model parameters on tables 7 and 8 seem speculative. The paper needs to better articulate how modeling the impact of attitudes towards interventions is better than the more straightforward approach of modeling different levels of coverage. Please clarify clearly in the introduction and discussion. Labeling of figures could be improved, several need labels for the Y axis. Using results from UNAIDS model to fit your model does not seem appropriate. Need to use prevalence data from surveys. Please review and address one by one comments from the two reviewers.

We look forward to receiving your revised manuscript.

Kind regards,

Gabriela Paz-Bailey

Academic Editor

PLOS ONE

Journal Requirements:

2.We note that you have indicated that data from this study are available upon request. PLOS only allows data to be available upon request if there are legal or ethical restrictions on sharing data publicly. For more information on unacceptable data access restrictions, please see http://journals.plos.org/plosone/s/data-availability#loc-unacceptable-data-access-restrictions.

3.Thank you for stating the following in the Acknowledgments Section of your manuscript:

[The authors thank the Organization for Women in Science for the Develop-

ing World (OWSD) for financing Ms. Ronoh's research visits to University

of KwaZulu-Natal (South Africa) where part of this research was done, the

Simons Foundation for meeting Ms. Ronoh's home institute (University

of Nairobi, Kenya) tuition costs]

 [The authors received no specific funding for this work]

Additional Editor Comments (if provided):

These analyses describe the trajectory of the HIV epidemic among youth in Kenya with various intervention coverages and with the added impact of positive or negative attitudes towards these interventions. Although attitudes can impact adherence to interventions, it is not clear how modeling attitudes and their impact in coverage is different than modeling the impact of varying coverage of the classical interventions (testing, ART, etc). Furthermore, it is challenging to measure how negative attitudes can affect level of coverage, so the model parameters on tables 7 and 8 seem speculative. The paper needs to better articulate how modeling the impact of attitudes towards interventions is better than the more straightforward approach of modeling different levels of coverage. Please clarify clearly in the introduction and discussion. Labeling of figures could be improved, several need labels for the Y axis. Using results from UNAIDS model to fit your model does not seem appropriate. You may be reproducing biases from the UNAIDS model, there must be prevalence data from surveys to evaluate the model fit.

Reviewers' comments:

Reviewer's Responses to Questions

**Comments to the Author**

1. Is the manuscript technically sound, and do the data support the conclusions?

Reviewer #1: Yes

Reviewer #2: Yes

2. Has the statistical analysis been performed appropriately and rigorously? 

Reviewer #1: Yes

Reviewer #2: I Don't Know

3. Have the authors made all data underlying the findings in their manuscript fully available?

Reviewer #1: Yes

Reviewer #2: Yes

4. Is the manuscript presented in an intelligible fashion and written in standard English?

Reviewer #1: Yes

Reviewer #2: Yes

5. Review Comments to the Author

Reviewer #1: Review of:

Evidence-based modeling of combinatory control on Kenyan youth HIV/AIDS dynamics by Ronoh et al.

The paper employs a deterministic model to study the effects of varying HIV/AIDS testing rates, condom use rates and ART adherence rates among Adolescent Girls and Young Women (AGYW) and, Adolescent Boys and Young Men (ABYM) populations in Kenya. The model is stratified by gender and divided into six mutually exclusive classes.

I find this paper very novel and well written and can be considered for publication. The scientific quality of this paper is high and coupled with the clarity of expression. However, the authors should attend to the following specific comments.

1. The authors should refrain from the use of time bound words like “recently”, “in the recent years”, etc., This will require contextualizing time in order to make sense. The authors should be more explicit and clear by quoting exact time/period by saying e.g., as of 2018, …. or Instead in July 2018, ….

2. In line 4, first paragraph, the authors should define what constitute “youth population” and in line 36 third paragraph “young adults”.

3. In line 48, “World Health organization” should be “World Health Organization”. Authors should correct such mistakes and all the typographical errors through the document.

4. The parameter estimations need a more careful analysis and discussion. The results of the fit should be discussed, and confidence intervals for estimates should be given.

5. The authors should place more emphasis on the novelty and importance of the results.

In summary, the manuscript does make a clear contribution and the results are novel and interesting to readers.

Reviewer #2: The article is an interesting modeling study that demonstrates the effects of different coverage levels of HIV control efforts and attitudes toward condom use, HIV testing and ART on the prevalence of HIV among AGYW and ABYM. While the main messages of the article are clear, the results seem somewhat absent or muddled, dedicating little real estate to the quantitative prevalence estimates predicted into the future.

I am not a mathematical modeler and cannot critique the minutiae of the mathematical model used. That said, I was surprised that the authors used UNAIDS' modeled prevalence estimates for AGYW and ABYM as a "truth" rather than the surveillance estimates that were used to parameterize the UNAIDS model. If every model is only as good as the inputs used, the model developed for this paper could be amplifying incorrect estimates by fitting to the UNAIDS yearly estimates. I recognize that the objective of this paper is not to replace the UNAIDS estimates, but it was the first time I had seen a model fitted to the estimates from another model.

Methods:

It was unclear from the methods section whether self-reported HIV status from the KAIS was taken at face value or adjusted using ARV metabolite or viral load testing. Survey participants often do not feel comfortable disclosing their HIV status to the interviewer and will report being HIV negative despite being on ART. The self-reported HIV status of people who are virally suppressed or who have ARV metabolites detected in blood samples can be used to correct/adjust the "awareness" of those infected.

Figures 1-3 were difficult to interpret given the lack of percentages or ratios comparing the distinct populations. The description of these figures interpreted the data from the KAIS as if it should have been in the results section of the paper but were used only to set up the model. Consider limiting the interpretation of these graphs as it doesn't tie into the main message of the paper and could be a separate paper if paid due diligence in the analysis. The use of the term "way higher" read as colloquial rather than scientific writing.

The variables (a, c, t) used to represent the different control measures are used in the figures which makes them more difficult to interpret. Specifying "condom use," "ART adherence" with the value for each figure would make the article more digestible.

Language: This may be a difference between American vs. British English but the use of "control(s)" was confusing. It took reading the entire article to understand that the authors meant "control measures/efforts/interventions/etc." Unless "controls is well understood broadly, consider revising to be more explicit.

See attached PDF with specific comments.

6. PLOS authors have the option to publish the peer review history of their article (what does this mean?). If published, this will include your full peer review and any attached files.

Reviewer #1: No

Reviewer #2: No

---

## [Author Response · Author response to Decision Letter 0]

19 Oct 2020

Manuscript Number: PONE-D-20-04014

Manuscript Title: Evidence-based modeling of combinatory control on Kenyan youth HIV/AIDS dynamics

RESPONSE TO REVIEWERS COMMENTS

The authors would like to thank all the Reviewers for their comments. We believe we have addressed them sufficiently. Kindly find the responses to all the comments below. 

Additional Editor Comments (if provided):

Comment

These analyses describe the trajectory of the HIV epidemic among youth in Kenya with various intervention coverages and with the added impact of positive or negative attitudes towards these interventions. Although attitudes can impact adherence to interventions, it is not clear how modeling attitudes and their impact in coverage is different than modeling the impact of varying coverage of the classical interventions (testing, ART, etc). Furthermore, it is challenging to measure how negative attitudes can affect level of coverage, so the model parameters on tables 7 and 8 seem speculative. The paper needs to better articulate how modeling the impact of attitudes towards interventions is better than the more straightforward approach of modeling different levels of coverage. Please clarify clearly in the introduction and discussion. 

Response

In Kenya, changing key HIV/AIDS control measures among the adolescents and young adults like HIV testing, condom use and ART adherence has faced significant challenges mostly due to societal attitudes towards the uptake of these controls by the youth. There is significant disparity in societal attitudes by gender towards the youth using some of these HIV/AIDS controls. On one hand, community norms and structural barriers directly affect condom use among the youth in Kenya. On the other hand, HIV knowledge, HIV-related stigma, income and social support from family and religious affiliations, mental health (depression, anxiety, stress) and substance use directly affect HIV test-seeking and treatment adherence among the youth. Given the direct link between societal attitudes and the uptake of the HIV/AIDS control measures among the youth, our study then highlights how these positive/negative attitudes influence the uptake of HIV/AIDS control measures. In this study, the positive/negative attitudes towards the use of HIV/AIDS control measures are designed to allow HIV testing, condom use and ART adherence to change over time. These social drivers directly influencing HIV testing, condom use and ART adherence are rarely addressed in mathematical modelling. Kindly refer to lines 61-72 and section 4 of the revised manuscript.

Comment

Labeling of figures could be improved, several need labels for the Y axis. 

Response

We did improve the figures that needed labels for the Y-axis. Kindly see figures 1-4 and 6 -14 in the revised manuscript.

Comment

Using results from UNAIDS model to fit your model does not seem appropriate. You may be reproducing biases from the UNAIDS model, there must be prevalence data from surveys to evaluate the model fit.

Response

Kenya has conducted five national surveys that have estimated HIV prevalence estimates for the adolescents and adult population namely; 2003 Kenya Demographic and Health Survey, 2007 Kenya AIDS Indicator Survey, 2008-09 Kenya Demographic and Health Survey, 2012 Kenya AIDS Indicator Survey and 2014 Kenya Demographic and Health Survey). In Kenya, the earliest case of HIV/AIDS was reported in the late 1980’s and it increased to a peak prevalence of 10.5% in 1995-96. The HIV prevalence estimates from these national surveys only capture part of the disease decline but fails to capture the initial rise to peak prevalence of the disease. The government of Kenya through the National AIDS Control Council in Kenya collaborates with Avenir Health, UNAIDS, public health professionals, demographers, global epidemiologists and monitoring and evaluation experts to annually provide Kenya's HIV/AIDS estimates. These experts use the Spectrum tools endorsed by UNAIDS to provide these estimates which are based on data from the five national surveys and, data from HIV Sentinel Surveillance among pregnant women, national census and data from various programmes. Hence, Kenya's annual HIV/AIDS prevalence estimates provided by UNAIDS reflect the existing HIV epidemic in the country. For this reason, we use the UNAIDS-Kenya HIV Surveillance data on Kenyan youth prevalence to fit the model prevalence for AGYW and ABYM populations. A similar study was conducted in South Africa where the mathematical model was fitted to UNAIDS prevalence data to understand epidemic trends in South Africa (Kindly refer to reference number 43 in the revised manuscript). There have been few studies that have modelled HIV epidemic trends among the youth population. While the UNAIDS data may have its limitations and biases, it does provide the prevalence trend of HIV/AIDS from its initial rise, peak prevalence and disease decline. This study does recommend more national surveys to be conducted so as to create a large incidence and prevalence data pool for finer model prevalence fit. Kindly refer to section 2.1.2 of the revised manuscript. 

REVIEWERS' COMMENTS:

Comments to the Author

1. Is the manuscript technically sound, and do the data support the conclusions?

Reviewer #1: Yes

Reviewer #2: Yes

2. Has the statistical analysis been performed appropriately and rigorously?

Reviewer #1: Yes

Reviewer #2: I Don't Know

3. Have the authors made all data underlying the findings in their manuscript fully available?

Reviewer #1: Yes

Reviewer #2: Yes

4. Is the manuscript presented in an intelligible fashion and written in standard English?

Reviewer #1: Yes

Reviewer #2: Yes

5. Review Comments to the Author

Reviewer #1: Review of:

Evidence-based modeling of combinatory control on Kenyan youth HIV/AIDS dynamics by Ronoh et al.

Comment

The paper employs a deterministic model to study the effects of varying HIV/AIDS testing rates, condom use rates and ART adherence rates among Adolescent Girls and Young Women (AGYW) and, Adolescent Boys and Young Men (ABYM) populations in Kenya. The model is stratified by gender and divided into six mutually exclusive classes.

I find this paper very novel and well written and can be considered for publication. The scientific quality of this paper is high and coupled with the clarity of expression. However, the authors should attend to the following specific comments.

Response 

Thank you very much for finding the study very novel and well written. We greatly appreciate that complement for it makes us want to do much better.

Comment

1. The authors should refrain from the use of time bound words like “recently”, “in the recent years”, etc., This will require contextualizing time in order to make sense. The authors should be more explicit and clear by quoting exact time/period by saying e.g., as of 2018, …. or Instead in July 2018, ….

Response

We refrained from the use of time bounds and made the necessary changes throughout the revised manuscript.

Comment

2. In line 4, first paragraph, the authors should define what constitute “youth population” and in line 36 third paragraph “young adults”.

Response

In this study, we define the youth as persons aged between 15 and 24 year based on the UN definition for youth. Kindly refer to the footnote in the first page of the revised manuscript. We replaced adolescents and young adults with youth. The revised manuscript reflects these changes. Kindly refer to lines 195-196 of the revised manuscript.

Comment

3. In line 48, “World Health organization” should be “World Health Organization”. Authors should correct such mistakes and all the typographical errors through the document.

Response

We corrected this (see line 46 of revised manuscript) and other typographical errors in the revised manuscript. 

Comment

4. The parameter estimations need a more careful analysis and discussion. The results of the fit should be discussed, and confidence intervals for estimates should be given.

Response

In our study, we used the ‘fminsearch’ algorithm in MATLAB to fit the model to data. This algorithm uses an alternative way to test the goodness of our model fit which we used instead of confidence intervals. We believe this is good enough to reach a similar conclusion if one uses the confidence intervals approach. Kindly refer to section 3.2 of the revised manuscript. The goodness of fit means that the model parameters can be estimated with little uncertainty (see reference 57 in the revised manuscript). There are many ways to test the goodness of our model fit and to show that the estimated parameters can be trusted. These include the sum of squares due to error (SSE), R-square, Adjusted R-square, root mean squared error (RMSE) and the confidence and prediction bounds. We used ‘fminsearch’ (which does not provide summary statistics) to fit our model to data using MATLAB software and obtained the minimum sum of squares due to error for the adolescents girls and young women model prevalence fit to be 0.0167 whereas the adolescents boys and young men was 0.0450. If the minimum value of the sum of squares due to error is closer to 0, it indicates that the model has a smaller random error component and the resulting fit is useful for prediction. Given the minimum SSE values we obtained were closer to 0, then the estimated parameters could then be trusted for the time series model simulations. Every algorithm for data fitting has its limitations but various indicators can be used to minimize errors and increase significance of the outcomes. Some studies (See [1] below) have also outlined the challenges with the use confidence intervals for infection transmission models since these models hardly have analytic methods for computing the confidence intervals and that the computed intervals are then viewed as prediction intervals rather than true confidence intervals. Confidence intervals methodology is designed for unimodal distributions, but numerical results of most infection transmission models are multimodal. In disease models, results outside the confidence intervals are important and these then cannot be observed when the approach of confidence intervals is used. In view of the arguments presented, the authors would love to maintain the analysis using the ‘fminsearch’ algorithm for it provided valid results suitable for the type of model and the predictions thereof. Kindly refer to section 3.2, lines 437- 446 of the revised manuscript. We highlighted studies that employed a similar routine of fitting their epidemic data to the model curve using ‘fminsearch’ algorithm. Kindly refer to references 58 – 60 in the revised manuscript.

Reference

[1]. E.T. Lofgren, Visualizing Results from Infection Transmission Models: A Case

Against “Confidence Intervals”, Epidemiology, 23(5), 738-741, 2012.

Comment

5. The authors should place more emphasis on the novelty and importance of the results.

Response

We addressed this comment in section 4 and expanded the discussion section to highlight the importance of our results. Kindly refer to section 4, paragraphs 2, 3 and 4 in the revised manuscript.

Comment

In summary, the manuscript does make a clear contribution and the results are novel and interesting to readers.

Response 

Thank you very much for outlining that the manuscript makes clear contribution and the results are novel and would be interesting to readers.

Reviewer #2: 

Comment

The article is an interesting modeling study that demonstrates the effects of different coverage levels of HIV control efforts and attitudes toward condom use, HIV testing and ART on the prevalence of HIV among AGYW and ABYM. While the main messages of the article are clear, the results seem somewhat absent or muddled, dedicating little real estate to the quantitative prevalence estimates predicted into the future.

Response

Section 3 in the revised manuscript reflects the results of the control reproduction number simulations, parameter estimation, data fitting, and the time series simulations. This now clearly establishes the results section. We provided prevalence estimates in section 3 lines 490 – 495. The results section captures both the qualitative and quantitative analysis of our study.

Comment

I am not a mathematical modeler and cannot critique the minutiae of the mathematical model used. That said, I was surprised that the authors used UNAIDS' modeled prevalence estimates for AGYW and ABYM as a "truth" rather than the surveillance estimates that were used to parameterize the UNAIDS model. If every model is only as good as the inputs used, the model developed for this paper could be amplifying incorrect estimates by fitting to the UNAIDS yearly estimates. I recognize that the objective of this paper is not to replace the UNAIDS estimates, but it was the first time I had seen a model fitted to the estimates from another model. 

Response

The government of Kenya through the National AIDS Control Council in Kenya partners with Avenir Health, UNAIDS, public health professionals, demographers, global epidemiologists and monitoring and evaluation experts to annually provide Kenya's HIV/AIDS estimates. These experts use the Spectrum tools endorsed by UNAIDS to provide these estimates which are based on data from the five national surveys ( 2003 Kenya Demographic and Health Survey, 2007 Kenya AIDS Indicator Survey, 2008/2009 Kenya Demographic and Health Survey, 2012 Kenya AIDS Indicator Survey and 2014 Kenya Demographic and Health Survey) and, data from HIV Sentinel Surveillance among pregnant women, national census and data from various programmes. Hence, Kenya's annual HIV/AIDS prevalence estimates provided by UNAIDS reflect the existing HIV epidemic in the country. For this reason, we use the UNAIDS-Kenya HIV Surveillance data on Kenyan youth prevalence to fit the model prevalence for AGYW and ABYM populations. A similar study was conducted in South Africa where the mathematical model was fitted to UNAIDS prevalence data to understand epidemic trends in South Africa (Kindly refer to reference number 43 in the revised manuscript). There have been few studies that have modelled HIV epidemic trends among the adolescents and young adults population. While the UNAIDS data may have its limitations and biases, it does provide the epidemic trend of HIV/AIDS for the youth population from its initial rise, peak prevalence and disease decline. Kindly refer to section 2.1.2 of the revised manuscript.

Comment

Methods:

It was unclear from the methods section whether self-reported HIV status from the KAIS was taken at face value or adjusted using ARV metabolite or viral load testing. Survey participants often do not feel comfortable disclosing their HIV status to the interviewer and will report being HIV negative despite being on ART. The self-reported HIV status of people who are virally suppressed or who have ARV metabolites detected in blood samples can be used to correct/adjust the "awareness" of those infected.

Response

The self-reported status referred to the respondents self-reported HIV status whereas KAIS confirmed HIV status referred to the respondents HIV status based on laboratory results from the survey. The KAIS confirmed HIV status took into account the viral load testing which we compared to the self-reported status thus adjusting the “awareness” of the uninfected and infected. In fact, we came across a scenario of respondents who were on ART and were KAIS confirmed negative HIV status. These respondents were previously tested elsewhere and erroneously received positive results. Hence, they were initiated on ART but later confirmed KAIS negative. Thus, we classified them as uninfected unaware. We have made this clarification in the methods section. Kindly refer to section 2.1.1 of the revised manuscript.

Comment

Figures 1-3 were difficult to interpret given the lack of percentages or ratios comparing the distinct populations. The description of these figures interpreted the data from the KAIS as if it should have been in the results section of the paper but were used only to set up the model. Consider limiting the interpretation of these graphs as it doesn't tie into the main message of the paper and could be a separate paper if paid due diligence in the analysis. The use of the term "way higher" read as colloquial rather than scientific writing.

Response

Figures 1-3 have percentages to compare distinct populations in the revised manuscript. We limited the interpretation of these graphs as advised. We only left the interpretations that justified the use of a sex structured model given that condom use, HIV testing and ART adherence were disproportional in the Kenyan youth population. Future studies may consider deeper analysis to expand this section to a separate paper. The term “way higher” was removed and replaced with facts. Kindly refer to section 2.1.1 of the revised manuscript.

Comment

The variables (a, c, t) used to represent the different control measures are used in the figures which makes them more difficult to interpret. Specifying "condom use," "ART adherence" with the value for each figure would make the article more digestible.

Response

We did make this change in all the figures that had the parameter symbols and specified them accordingly. Kindly refer to the figures in the revised manuscript.

Comment

Language: This may be a difference between American vs. British English but the use of "control(s)" was confusing. It took reading the entire article to understand that the authors meant "control measures/efforts/interventions/etc." Unless "controls is well understood broadly, consider revising to be more explicit.

Response

We revised controls to control measures in the revised manuscript. Kindly refer to the marked manuscript.

See attached PDF with specific comments.

(Comments from PDF extracted and responses given below)

Comment (Abstract) 

Unclear what is meant by controls. “control measures” like interventions or is this a modelling term?

Response

In the revised manuscript, we have used “control measures”. This change was made throughout the manuscript. Kindly refer to the marked manuscript.

Comment (Abstract) 

General results. Do you have space to be more specific regarding the prevalence trends that resulted from the model?

Response

We included a few lines in the abstract regarding the specific prevalence trends that resulted from the study. Kindly refer to the abstract section in the revised manuscript.

Comment (Line 117)

Do you have any way to account for people who may have been aware but did feel comfortable disclosing their status to the interviewer. For example people who are virally suppressed are most likely aware and on ART despite what they self-report as their HIV status.

Response

The case that we came across were adolescents and young adults who were on ART despite a negative HIV status. These were persons who were previously tested and received positive results and initiated on ART and were later confirmed KAIS negative. The respondents who reported negative were further tested in this survey. KAIS confirmed HIV status took into account the viral load testing which we compared to the self-reported status thus adjusting the “awareness” of the uninfected and infected. We clarified this in section 2.1.1 of the revised manuscript.

Comment (Line 121)

A ratio or comparison of percents of ABYM that were (un)infected and aware vs. unaware by sex would convey the message more clearly and precisely than these graphs of counts - the reader must estimate the difference for ABYM vs. AGYW.

Response

We did adopt this change. Kindly refer to lines 136 - 137 of the revised manuscript.

Comment (Line 127)

Could use adolescents and youth rather than dual acronyms which interrupt the flow of the sentence.

Response

We did make this change and used youth. Kindly refer to lines 142 and 144 of the revised manuscript.

Comment (Figure 2 Caption)

Percentages would make these graphs easier to interpret. I find myself looking from one column to another trying to guess the % that used a condom. The graphs as currently presented are not easy to interpret.

Response

We did include percentages in the revised graphs. Kindly refer to section 2.1 of the revised manuscript. 

Comment (Line 130 )

Avoid the term “way higher” as a colloquial term. Try considerably, noticeably, markedly, significantly (if using statistical testing) or other similar words.

Response

We did remove the term “way higher” and other similar words. These have now been replaced with factual figures. Kindly refer to section 2.1 of the revised manuscript.

Note (Lines 133-134 )

Highlighted but comment missing

Comment (Line 138)

Avoid the term “way higher” as a colloquial term. Try considerably, noticeably, markedly, significantly (if using statistical testing) or other similar words.

Response

We did remove the term “way higher” and other similar words. These have now been replaced with factual figures. Kindly refer to section 2.1.1 of the revised manuscript.

Comment (Figure 3)

If the fact that being unaware of HIV infection precludes being on ART, why are those unaware included in the figure? Again, percentages over counts would facilitate interpretation.

Response

Figure 3 reflects percentages and we did remove the unaware from the figures. Kindly refer to figure 3 in the revised manuscript.

Comment (Lines 170-171)

Where is the link between transactional sex and inclusion/exclusion criteria?

Response

There are studies that have shown that transactional sex could fuel HIV transmission among the youth. Given that this formulation was based on the 2012 KAIS data, it was not clear that transactional sex was a driving influence to HIV transmission hence, we excluded it from the formulation.

Comment (Lines 205-206 )

What are these probabilities?

Response

γ_f,γ_m are probabilities that a susceptible person coming into proper contact with an infected individual will contract the disease. Kindly refer to lines 228-230 of the revised manuscript.

Comment (Table 6 )

Are the attitude factors and interaction of these controls with the positive/negative attitudes designed to allow the controls to change over time as you would expect in reality?

Response

Yes. The attitudes considered are factors and interactions of HIV testing, condom use and ART adherence. Thus, the positive/negative attitudes are designed to allow these control measures to change over time as in the real world.

Comment (Line 435)

Why would the prevalence decrease in the absence of controls. I would expect the opposite as stated in the next sentence.

Response

We expect prevalence decrease in places where the disease is established. We did remove this statement in the revised manuscript so as to avoid confusion. Kindly refer to line 466 of the revised manuscript.

Comment (Figure 9)

Recommend aligning the y-axes and potentially starting at 0

Response

We did align the y-axis of figures 9, 10(b), 14(b) and 14(d) and they start from 0. Kindly refer to figure 9, 10(b), 14(b) and 14(d) in the revised manuscript.

Comment (Table 7)

Where did these estimates come from?

Response

We arrived at these estimates after fitting the model prevalence to prevalence data. We have clarified this in the manuscript. Kindly refer to lines 457- 458 and Table 7.

Comment (Figure 10)

Comment incomplete.

Comment (Lines 485-486)

Revise syntax

Response

We revised the statement. Kindly refer to lines 530 - 532 of the revised manuscript.

Comment (Figure 13)

Response: Figure 13 highlighted but comment missing.

Comment (Figure 14)

Are these the results of the model or Fig. 10? I would expect quantitative results of the modeling exercise to be presented in the Results section rather than the Methods section.

Response

Section 3 in the revised manuscript reflects the results of the control reproduction number simulations, parameter estimation, data fitting, and model simulations.

Comment (Line 584)

Combination prevention/control may be more familiar to readers

Response

We used combination control. This was applied everywhere where combinatory control appeared. Kindly refer to the marked manuscript.

Comment (Line 587)

Should be populations’

Response

We have corrected this. Kindly refer to line 633 of the revised manuscript.

Comment (Line 599)

Delete. Screening usually refers to HIV using rapid testing whereas AIDS is a clinical diagnosis

Response

AIDS deleted as advised. Kindly refer to line 680 of the revised manuscript. It now reads HIV screening.

Comment (Lines 601 and 602)

Control efforts/measures/programs/interventions

Response

We used control measures. We updated this in the entire manuscript. Kindly refer to the marked manuscript.

Comment (Lines 603 - 605)

Revise syntax

Response

We revised this statement. Kindly refer to lines 684 – 692 of the revised manuscript.

Comment (Line 619)

Delete

Response

Deleted AIDS disease as advised. Kindly refer to line 707 of the revised manuscript.

Comment (Line 620)

Delete. Only HIV infection is possible

Response

Deleted AIDS as advised. Kindly refer to line 708 of the revised manuscript.

---

## [Editor Report · Decision Letter 1]

4 Nov 2020

Evidence-based modeling of combination control on Kenyan youth HIV/AIDS dynamics

PONE-D-20-04014R1

Dear Dr. Ronoh,

We’re pleased to inform you that your manuscript has been judged scientifically suitable for publication and will be formally accepted for publication once it meets all outstanding technical requirements.

Kind regards,

Gabriela Paz-Bailey

Academic Editor

PLOS ONE

Additional Editor Comments (optional):

Thanks for submitting the revised manuscript. The article is accepted for publication.
---

## [Editor Report · Acceptance letter]

6 Nov 2020

PONE-D-20-04014R1 

Evidence-based modeling of combination control on Kenyan youth HIV/AIDS dynamics 

Dear Dr. Ronoh:

I'm pleased to inform you that your manuscript has been deemed suitable for publication in PLOS ONE. Congratulations! Your manuscript is now with our production department. 

Kind regards, 

on behalf of

Dr. Gabriela Paz-Bailey 

Academic Editor

PLOS ONE